# Unknown Extracellular and Bioactive Metabolites of the Genus *Alexandrium*: A Review of Overlooked Toxins

**DOI:** 10.3390/toxins13120905

**Published:** 2021-12-16

**Authors:** Marc Long, Bernd Krock, Justine Castrec, Urban Tillmann

**Affiliations:** 1IFREMER, Centre de Brest, DYNECO Pelagos, 29280 Plouzané, France; marc.florian.long@gmail.com; 2Alfred Wegener Institute for Polar and Marine Research, Am Handelshafen 12, 27570 Bremerhaven, Germany; Bernd.Krock@awi.de; 3University Brest, CNRS, IRD, Ifremer, LEMAR, 29280 Plouzané, France; Justine.Castrec@univ-brest.fr; 4Station de Recherches Sous-Marines et Océanographiques (STARESO), Punta Revellata, BP33, 20260 Calvi, France

**Keywords:** dinoflagellate, paralytic shellfish toxin, lytic, allelopathy, bioactivity, chemical ecology, secondary metabolite

## Abstract

Various species of *Alexandrium* can produce a number of bioactive compounds, e.g., paralytic shellfish toxins (PSTs), spirolides, gymnodimines, goniodomins, and also uncharacterised bioactive extracellular compounds (BECs). The latter metabolites are released into the environment and affect a large range of organisms (from protists to fishes and mammalian cell lines). These compounds mediate allelochemical interactions, have anti-grazing and anti-parasitic activities, and have a potentially strong structuring role for the dynamic of *Alexandrium* blooms. In many studies evaluating the effects of *Alexandrium* on marine organisms, only the classical toxins were reported and the involvement of BECs was not considered. A lack of information on the presence/absence of BECs in experimental strains is likely the cause of contrasting results in the literature that render impossible a distinction between PSTs and BECs effects. We review the knowledge on *Alexandrium* BEC, (i.e., producing species, target cells, physiological effects, detection methods and molecular candidates). Overall, we highlight the need to identify the nature of *Alexandrium* BECs and urge further research on the chemical interactions according to their ecological importance in the planktonic chemical warfare and due to their potential collateral damage to a wide range of organisms.

## 1. Introduction

### 1.1. Alexandrium, a Potentially Toxic Genus

Exceptional densities of marine microalgae, commonly reported as blooms, are recurrently observed in many coastal areas around the world. A number of dinophyceae microalgae are producers of potent phycotoxins which, during such blooms, may have major economic (e.g., on tourism or exploitation of marine resources) and/or health impacts (e.g., human poisoning). Among toxigenic and bloom-forming dinophytes, the genus *Alexandrium* Halim is perhaps the most intensely studied. This is largely related to the ability of several *Alexandrium* species to produce paralytic shellfish toxins (PSTs; saxitoxin and analogues) responsible for life-threatening human poisoning (paralytic shellfish poisoning, PSP) trough consumption of contaminated seafood [1]. Other species of *Alexandrium* may produce other phycotoxins such as the spirolides and/or gymnodimines produced by *A. ostenfeldii* (Paulsen) Balech and Tangen (reported as *A. peruvianum* by [2]), or goniodomins produced by a few other *Alexandrium* species [3]. Moreover, many species are described to produce poorly characterised extracellular compounds with lytic capacity [4]. All these traits may contribute to the devastating consequences of toxic *Alexandrium* blooms including human poisoning, fish kills, socio-economic losses to aquaculture and fisheries, marine fauna mortality and food web disruptions [5].

### 1.2. Ecology of Alexandrium

Blooms of *Alexandrium* are favoured by suitable biotic and abiotic environmental conditions and may be triggered and enhanced by increased water temperature, high irradiance, high nutrient supply or water stratification [6,7,8]. Blooms of *Alexandrium* can be of high biomass with maximum cell densities up to 10^7^–10^8^ cells L^−1^ [9,10], and at times can be almost monospecific [6,11,12,13,14]. Blooms may occur as both large scale coastal events [15,16,17,18] as well as regional events in estuaries and coastal embayments [19,20] and are often associated with substantial economic losses due to the closure of shellfish beds or even mortality of shellfish [17]. The success and dominance of *Alexandrium* species in plankton communities and the ability for blooms to persist and to attain high cell densities highlight their well-developed adaptative capacities including competition and/or defensive mechanisms against biotic pressures (e.g., competitors, grazers and parasites). All species of *Alexandrium* have typical peridinin plastids and are phototrophic. A number of species, however, have been shown to be mixotrophic [21,22], and this trophic versatility may also contribute to bloom formation. Another important factor of recurrent blooms of *Alexandrium* are cysts beds. As part of their life cycle, many species of *Alexandrium* can form benthic cysts (in most cases hypnozygotes), that allow cells in dormancy to stand unfavourable temperature or nutrient conditions [23]. Subsequent cyst germination can occur during suitable growth conditions and inoculate vegetative cells into the water column. This can thus initiate blooms, the extent of which may depend on cyst bed distribution as well as on abundance of cysts formed at the end of previous vegetative periods [23,24]. Cysts are also critical in species dispersal, as cells transported to new locations by storms, currents, or humans (e.g., in ballast water), can colonise an area by depositing cysts that can germinate in the subsequent years [24].

### 1.3. Taxonomy and Nomenclature of Alexandrium

*Alexandrium,* a typical gonyaulacoid genus in the subfamily Ostreopsidoideae [25] was erected by Halim [26]. The taxonomic history of the genus is quite complex and include numerous rearrangements of species formerly classified in *Gonyaulax* Diesing, *Protogonyaulax* F.J.R.Taylor, *Gessnerium* Halim, *Goniodoma* F.Stein, and *Pyrodinium* L.Plate [5,27]. The very first species assignable to *Alexandrium* was described by Paulsen back in 1904 as *Goniodoma ostenfeldii* Paulsen (=*Alexandrium ostenfeldii* (Paulsen) Balech and Tangen) from plankton samples from Iceland [28], followed by the description of *Gonyaulax tamarensis* Lebour (=*Alexandrium tamarense* (Lebour) Balech) from the Tamar estuary (southern England) in 1925 [29], and the description of *Gonyaulax catenella* Whedon and Kofoid (=*Alexandrium catenella* (Whedon and Kofoid) Balech) from the north-western Pacific Ocean off San Diego [30]. After rearranging these and other species into *Gessnerium* [31] and/or *Protogonyaulax* [32], a scientific consensus for the taxonomy of these species was finally reached when Balech [33,34] redefined Halim’s genus *Alexandrium* and transferred a total of 22 species into *Alexandrium*. Enrique Balech (1912–2007), an eminent taxonomist from Argentina, greatly impacted our knowledge on *Alexandrium* by describing as much as eleven new *Alexandrium* species and by transferring many other species into *Alexandrium*. His monograph on the genus [35], where he compiled all available information on how to identify and differentiate the species, is still the benchmark in the field. Despite the thorough work of Balech, a number of new *Alexandrium* species have been discovered in the 20th century. In 2004, *A. tamutum* M. Montresor, U. John and A. Beran, *A. gaarderae* L. Nguyen-Ngoc and J. Larsen, and *A. globosa* L. Nguen-Ngoc and J. Larsen were added, followed by *A. diversaporum* S.Murray et al., *A. pohangense* A.S. Lim and H.J. Jeong, and *A. fragae* S. Branco and M. Menezes [36,37,38,39,40]. All these new species, as far as tested with cultured strains, are not producing PSTs. Moreover, in 2020, a molecular characterisation of multiple strains isolated from marine macroalgae revealed three novel phylotypes of predominantly coccoid live stages. The phylotypes nested in *Alexandrium* but low availability of motile cells yet prevented a thecal plate analysis and a formal species description and diagnosis [41]. Finally, with the availability of first sequence data of the fusiform gonyaulacoid genus *Centrodinium* Kofoid [42] it became clear that at least three of their species (*C. punctatum* (Cleve) Taylor, *C. eminens* Böhm, *C. intermedium* Pavillard) form a clade nesting within *Alexandrium* [42,43]. Moreover, it was also shown that *C. punctatum* produces exceptionally large amounts of PSP toxins [44]. Notably, the phylogenetic nesting of *Centrodinium* in *Alexandrium* makes the latter paraphyletic. The nomenclatural consequences are not clear yet, especially as morphological and molecular details on the type species of *Centrodinium*, *C. elongatum* Kofoid, are not available. Recently, Gómez and Artigas [43] in an attempt to “solve” *Alexandrium*´s paraphyly, proposed to split *Alexandrium* into four genera (retaining *Alexandrium*, resurrecting *Gessnerium* and *Protogonyaulax*, and newly erecting *Episemicolon*). However, as argued by Mertens and collaborators [45], these reintroduced taxa were not based on monophyletic groups, and accepting the Gómez and Artigas proposal would result in replacing a single paraphyletic taxon with several non-monophyletic ones so that the proposal [43] to split *Alexandrium,* on the basis of our current knowledge, should be rejected.

The general problem of reliable species identification in the protistan realm especially refers to *Alexandrium*, where most taxa are rather similar in general size and shape [35] but where the presence of toxigenic species calls for a sound and reliable species determination. Identification of *Alexandrium* species is not a simple task and requires a thorough examination of morphological differences in cell size, shape, or chain formation. Most importantly, morphological species identification requires to reveal a suite of subtle details of the theca such as ornamentation, presence and development of cingular and sulcal lists and excavations, the presence and location of attachment pores and of a ventral pore, and shape and arrangement of diagnostic thecal plates such as the apical pore plate, the first apical plate, or the sulcal plates [1,35]. While advanced molecular techniques are of increasing value to support and to ease species identification, a number of phylogenetic studies related to *Alexandrium* revealed cryptic speciation and also invalidated some of the described morphospecies [46,47]. This is especially important as it particularly concerns the most important PSTs producing species complexes of *A. minutum* Halim and the former *A. tamarense/catenella/fundyense* species complex.

Within the *A. minutum* group *sensu* Balech (1995), several morphospecies have been described (*A. minutum*, *A. ibericum* Balech, *A. angustitabulatum* F.J.R.Taylor, and *A. lusitanicum* Balech) with rather similar morphological characteristics and almost identical PSP toxin profiles. A detailed molecular ribosomal gene sequence analysis of members of the *A. minutum* group convincingly revealed that *A. lusitanicum* and *A. angustitabulatum* are in fact synonymous with *A. minutum*, whereas other morphologically similar species such as *A. tamutum*, *A. insuetum*, and *A. andersonii* were confirmed to be valid species [46].

The most prominent example of an *Alexandrium* morphospecies concept failure for species circumscriptions is the former *A. tamarense* species complex consisting of the morphospecies *A. tamarense*, *A. catenella*, and *A. fundyense* Balech [47,48,49]. For several decades these morphospecies were reported and named in the literature based on whether cells were anterior-posteriorly compressed and formed chains (“*A. catenella*” morphospecies) or not (“*A. tamarense*” and “*A. fundyense*” morphospecies) and based on the presence (“*A. tamarense*” morphospecies) or absence (“*A. fundyense*” morphospecies) of a ventral pore. However, the first molecular phylogenetic studies of RNA sequences obtained from numerous strains from globally distributed localities of the species complex revealed five distinct ribotypes (named groups I to V) which however did not conform with the morphological criteria [50]. Consequently, in a detailed study including morphology, molecular phylogeny based on multiple marker genes, mating compatibility and presence/absence of STX-coding genes, the five ribotypes where described as distinct species, i.e., *A. catenella* for group I, *A. mediterraneum* U.John for group II, *A. tamarense* for group III, *A. pacificum* R.W.Litaker for group VI, and *A. australiense* Sh.Murray for group V [48,51]. These nomenclatural changes have important implications [52]. While it is clear that current and future identification of *Alexandrium* species of this group has to be based on rRNA marker gene sequencing and/or use of species–specific molecular assays, the use of previous paper information to compile and/or compare species-based information is challenging. A strain previously reported as “*A. tamarense*” does not necessarily (and in fact in many cases probably does not) provide trait description for the species *A. tamarense* as it is described today. There is no doubt that species can/do differ in trait and thus reliable species identification and an unambiguous use of scientific names is an indispensable base for any communication about biological species and their traits. In the following review we thus report and compile information of older papers as follows: (1) if the previous work using strains of the *tamarense/catenella/fundyense* species complex provide molecular data or a strain identifier for which molecular data are available elsewhere, we report the results under the new correct name, followed by the name reported in that paper in parenthesis (e.g., *A. catenella* (reported as *A. tamarense*)). As a comprehensive compilation of strain identifiers linked with molecular data, we used the supplementary file S9 provided in [48]. When we were unable to locate/identify sequence data for a particular strain reported in the literature, we reported the species name using quotation marks and adding (not further characterised) (e.g., “*A. tamarense*” (not further characterised). In some cases, where additional data of the respective paper (e.g., geographical origin, presence/absence of PSTs) are available, this information was included in the parentheses (e.g., “*A. tamarense*” (likely to represent *A. catenella* as the strain produce PSTs)). For all other species names of *Alexandrium*, we conformed to synonymies as reported in AlgaeBase [53] and corrected species names from previous papers accordingly.

### 1.4. Effects of Alexandrium on Marine Biota

Because of the economic importance of PST-producing species, direct effects of *Alexandrium* on marine biota have long been studied, especially on the physiology of economically important shellfish, but also on finfish and commercial fisheries. Deleterious physiological effects and marine biota mortality events have been associated with *Alexandrium* outbreaks [17,54,55,56,57]. So far, these deleterious effects have largely been attributed to the neurotoxic PSTs. But the general problem is that PSTs are not the only bioactive compounds produced by *Alexandrium*, which makes it difficult or often impossible to pinpoint the ultimate cause of these negative effects. As will be discussed in detail below, many, if not all, species of *Alexandrium* are known to release bioactive extracellular compounds (BECs) that induce deleterious (mostly lytic) effects against a wide range of cells from protists to mammalian cell lines. In the plankton realm, BECs may negatively affect competing protists [58,59,60], immobilise prey [61,62] and predators [4,63], inhibit putative parasites [64], or affect grazing of copepods [65]. Moreover, there is growing evidence that BECs might also be responsible for physiological incapacitation of shellfish [66,67,68,69,70,71,72] or fish [73]. Nevertheless, detailed knowledge on chemical details of the metabolites mediating these interactions is largely lacking. Most of the toxic effects of *Alexandrium* on commercial species have been attributed to the known toxins while BECs were not investigated. The objective of this review is to summarise and merge the current knowledge from different fields on BECs within the genus *Alexandrium*.

## 2. Known Toxins of *Alexandrium* and Their Effects

An extraordinary characteristic of *Alexandrium* is the diversity of toxins that a number of species produce (Figure 1, Table 1). This does not only refer to the chemical variability of a toxin and its derivatives, but also includes the presence of chemically very different toxin classes. Currently, the known toxins associated with species of *Alexandrium* are hydrophilic paralytic shellfish toxins (PST), and various lipophilic compounds such as spirolides (SPX), gymnodimines (GYM), and goniodomins (GD).

### 2.1. Paralytic Shellfish Toxins

The best studied group of *Alexandrium* toxins are PSTs. This probably is due to their rapid accumulation in filter-feeding bivalves and their high toxicity in humans after consumption of contaminated shellfish [74]. PSTs are also associated with deleterious effects in seabirds [75] and marine mammals [56,76]. PSTs were already reported in the first half of the 20th century [77] and comprise a group of over 40 variants, not only produced by *Alexandrium* (including *Centrodinium punctatum,* which cluster within *Alexandrium*) but also by two other species of different genera (*Gymnodinium catenatum* H.W.Graham and *Pyrodinium bahamense* L.Plate), as well as several species of different genera of freshwater cyanobacteria. The PSTs produced by *Alexandrium* are constrained to 12 structural variants namely carbamoyl and *N*-sulfocarbamoyl toxins. The composition of these variants is a stable phenotypic trait, but the amounts of PSTs derivatives vary between and within strains [1]. In vertebrates including humans, PST target voltage-gated sodium channels inducing a blockage of ion transport and action potential of excitable membranes and are accordingly classified as neurotoxins. PSTs accumulation and shellfish contamination may be responsible for PSP outbreaks, thus leading to economic losses linked with fishery closures or recreational activities, and human intoxications. Moreover, PSTs have been associated with ichthyotoxicity [78,79], toxicity in shellfish [80], and deterrence or toxicity to copepods [81,82].

### 2.2. Cycloimines (Spirolides and Gymnodimines)

Cycloimines are a group of marine biotoxins characterised by a macrocyclic carbon skeleton and an additional six or seven membered rings with an imine functional group [83]. Within this group, spirolides are characterised by a seven membered cyclic imine moiety, which were first discovered in shellfish, because of their fast neurotoxic properties (so-called fast-acting toxins) detected in the mouse bioassay [84]. Later, spirolides were reported to be produced by *Alexandrium ostenfeldii* [85], the only known species producing spirolides. In recent years, increasingly more spirolide variants were detected in *A. ostenfeldii* from different geographic regions [2,86,87,88,89,90,91,92,93,94,95,96,97]. Chemically related gymnodimines were originally detected in shellfish in New Zealand [98] and soon associated with the naked marine dinoflagellate *Karenia selliformis* A.J. Haywood, K. Steidinger and L. MacKenzie (initially reported as *Gymnodinium* sp.) [99]. Later, 12-methyl gymnodimine was detected in *A. ostenfeldii* (reported as *A. peruvianum* (Balech and Mendiola) Balech and Tangen) [2] and since then many other gymnodimine variants were found in this species [87,89,100].

Despite their very high intraperitoneal toxicity, cycloimines are far less toxic when administered orally [101]. Functional bioassays on gymnodimine A (GYM A) and 13-desmethyl spirolide C (SPX 1) revealed a similar bioactivity of these toxins as both GYM A and SPX 1 induced rapid neurotoxic symptoms in mice, suggesting that both are fast-acting toxins [102]. The common symptoms induced by GYM A and SPX 1 via different routes were proven to be caused by binding of cycloimines to acetylcholine receptors (AChRs) [103,104]. Electrophysiological inhibition assays on GYM A [105] and SPX 1 [106] described the capacity of the toxins to inhibit AChRs. However, a reversible effect was only apparent in treatments with GYM A [105] indicating slightly different modes of action of these chemically very similar compounds. This was further supported by Nieva and collaborators, who showed that GYM A and SPX 1 activated nicotinic AChRs, but that only GYM A activated muscarinic AChRs [107].

### 2.3. Goniodomins

The first report of goniodomin (GD) dates back to 1968 in Puerto Rico, when *Goniodoma* sp. (later specified as *G. pseudogonyaulax* Biecheler and subsequently revised as the new species *A. hiranoi* Kita and Fukuyo [108]) was described as a source of GD [109]. At that time the structure of this GD was not fully elucidated, but twenty years later the compound was isolated again from a bloom of *A. hiranoi* and described as the macrocyclic polyketide goniodomin A (GDA) [110]. Recent compelling evidence shows that goniodomin and GDA in fact are the same compound [111]. Later reports added that GDA is also produced by *A. monilatum* (J.F.Howell) Balech [112], *A. pseudogonyaulax* (Biecheler) Horiguchi ex K.Yuki and Y.Fukuyo [113], and *A. taylorii* Balech [3]. In 2008 an isomer of GDA was structurally elucidated and named GDB [114]. Already Sharma and collaborators [109] described an antibiotic activity of GD and later other biological effects have been added, such as antifungal activity [110], cell division inhibition of sea urchin eggs [110], antiangiogenic activity [115], and perhaps most significantly in terms of its adverse effects on marine ecosystems, it has been associated with mortality in aquatic invertebrates [116]. Not much is known about the transfer of GDA in food webs, but GDA has been shown to accumulate in the marine snail *Rapana venosa* in controlled exposure experiments [116]. Cytotoxicity and effects on actin levels in human neuroblastoma cells of GDA and GDB were reported [117]. Moreover, it was shown that GDA forms potassium (and other alkali ion) complexes with potentially ionophoric properties, which may be involved in allelochemical interactions and ichthyotoxicity [118].
toxins-13-00905-t001_Table 1Table 1Compilation of toxin information and trophy of *Alexandrium* species. No published information was found for the following species: *A. balechii*, *A, compressum*, *A. camurascutulum*, *A. concavum*, *A. depressum*, *A. foedum*, *A. fraterculus*, *A. gaarderae*, *A. globosum*, *A. kutnerae*, *A. satoanum*, *A. tropicale*, and *A. acatenella*. Detailed information on BECs activity is available in Table 2. In the table, “+” indicates that the vast majority of strain of the species produce toxins, “−” indicates that in the cited reference, the presence of the compound was specifically looked for but it was not detected. Presence of “±” indicates that both toxigenic and non-toxigenic strains of the species have been described or that contrasting results about toxin production exist. Absence of symbol indicates that no information is available from the literature. All species of *Alexandrium* have plastids and are photosynthetic. For a couple of species mixotrophic capability have been tested, and these species are marked here as “P” (i.e., phototroph, phagotrophy was not observed (but note that always a limited number of different strains has been tested) or “M” (i.e., mixotroph, phagotrophy was observed).SpeciesPSPSpiro-ImineGonodiominTrophy*A. affine*± ^a^

P [21]*A. andersonii*± ^b^

M [21]*A. australiense*± ^c^


*A. catenella*+ ^d^− ^e^
M [119]*A. cohorticula*± ^f^


*A. concavum*− ^g^


*A. diversaporum*− ^h^


*A. fragae*+ ^i^


*A. fraterculus*− ^j^

P [21]*A. hiranoi*

+ ^k^P [120] *A. insuetum*− ^l^

P [121] *A. leei*± ^m^


*A. margalefii*− ^n^

P [120] *A. mediterraneum*− ^o^

P [121] *A. minutum*± ^p^

M [119] *A. monilatum*− ^q^

+ ^r^
*A. ostenfeldii*± ^s^± ^s^
M [122]*A. pacificum*+ ^t^

P [121] *A. pohangense*− ^u^

M [61]*A. pseudogonyaulax*− ^v^
+ ^v^M [120]*A. tamarense*− ^w^

M [123]*A. tamiyavanichii*+ ^x^


*A. tamutum*− ^y^

P [121] *A. taylorii*− ^z^− ^z^+ ^z^P [120](a) Negative for PST in [124], the only positive record comes from [125]. HPLC estimates, but without any details on methods. (b) PSTs reported in [126], but: negative for PSP [127], also negative for PST [124]. Negative for PSP also in [128] (but no limit of detection (LOD) reported), negative for PSP [129] (including a lack of STX genes). (c) PSTs and stxA confirmation but for only one strain in [130]. Species otherwise considered to be non-PST [131,132]. (d) Characterisation of PSTs production in [133]. (e) Lack of spirolides reported in [134] for strains of “*A. tamarense*” likely to be *A. catenella* (no LOD given). (f) Comment in [1]: Japanese strains reportedly toxigenic, but possible misidentification of *A. tamiyavanichii*. (g) Lack of PSP in [135] but their strain designated as *A. gaarderae* is not *A. gaarderae* as described by Gaarder but corresponds to *A. gaarderae* as defined in [39]. (h) Lack of PSTs and lack of sxtA documented in [36]. (i) PSP in [38]. (j) Lack of PST in [136], also in [135], also in [124]. (k) Goniodomin A was isolated from a bloom of *A. hiranoi* (the strain was initially described as *Goniodoma pseudogonyaulax* [110]). (l) Lack of PSP in [137] (in Japanese, also in [124]. (m) Comment in [1]: “Typically non-toxic, but low level of saxitoxin derivative reported from Vietnamese strain”. The positive record comes from [125] but no details (e.g., no strain identifier, no sequence data, and no morphology) on *A. leei* given and very low levels or even no toxins (but without LODs). Negative record for PSP in [138] (but only receptor assay, no LOD given). (n) Lack of PSP in [135], also in [128] (no LOD reported). (o) Claimed as no PSP [48]. (p) PST producing species [139,140] but non-PSP strains reported in [141,142]. (q) Lack of saxitoxin in *A. monilatum* extracts and different mode of toxicity in [143,144] (r) Goniodomin A reported in [112]. (s) Strains can be PST or non-PST producers [1]. The strain variability in spiroimines refers to gymnodimines. Spirolides seem to be produced by all strains [145]. There is one report of a strain from Chile [146] reporting lack of spirolides, but this needs re-investigation, as recent analyses indicate that this strain may produce previously unknown spirolides (Krock, unpublished). (t) PST production characterised in [140]. (u) Absence of sxtA documented in [37]. (v) Lack of PST documented in [135,147], also [128] (no LOD reported). Goniodomin A and B reported in some strains [147]. (w) Claimed as no PSP [48]. A PSTs producing strain of *A. tamarense* was recently reported from the Mediterranean [148], but, as stated by the authors, toxin analysis of this strain was carried out at the beginning of the 2000s, and toxin data of Mediterranean *A. tamarense* strains should be confirmed in the future using more recent and proven methods and instruments. (x) Presence of sxtA documented in [149]; PSP analysis in [124,150,151,152]. (y) Negative for PSP in species description paper [40]. Confirmed negative for PSP by [141,153,154], negative also in [127,128] (but no details on LOD). (z) Mediterranean strains of *A. taylorii* do not produce PSP [3,4] nor spiroimines, but do produce Goniodomin A [3]. A Pacific strain designated as *A. taylorii* without accompanying sequence data has been claimed as a PSP producer [155], but this report is questionable and needs re-investigation: the cell quotas were very low and the “toxin profile” had a 100% match with the PSP profile of a *A. ostenfeldii* strain which was simultaneously studied, and thus strain cross-contamination and not PSP production of *A. taylorii* is the likely cause of this report.

## 3. Uncharacterised Extracellular Bioactivities

Species of *Alexandrium* not only produce the known toxins listed above, but also uncharacterised bioactive extracellular compounds (BECs), which at present are mainly described by their lytic activity against a wide range of organisms. Lytic activity has been studied with standard haemolytic assays or with protistan competitors or predators as target species. However, lysis has also been reported against other cell types, such as para-sites, fish gill cells and shellfish tissue (Table 2).

Effects of BECs have been reported for many species of *Alexandrium* (Table 2). A high variability in the potency is observed among strains of the same species [4,156,157], even within strain from the same population [158,159,160,161].

There is experimental evidence that BECs activities are not mediated by PSTs or cycloimines (including gymnodimines and spirolides) but by other compounds, which we qualify as BECs [4,162,163]. These metabolites seem of dinoflagellate origin as allelochemical potency persisted when bacterial load was drastically decreased by using antibiotics [58,164,165]. However, full axenia may not have been reached and, therefore, it cannot be excluded that bacteria may play a role in the potency of BECs. For *Alexandrium* monocultures grown in the laboratory, there is a constitutive release of BECs in both exponential and stationary growth phases [157,166]. The production of BECs, like the production of any organic molecule, has an energetic cost, but the implications for the production of BECs is not clear yet. Preliminary comparison of a number of *Alexandrium* strains indicates that, to some extent, lytic activity and growth rate are inversely correlated, but only when PST-producing strains are included and non-lytic strains are excluded [167]. Another study [168] evidenced a trade-off between lytic potency and feeding ability (i.e., mixotrophic behaviour) of one strain of *A. pseudogonyaulax*, thus supporting the idea that BECs impose a significant energetic cost on the cell.

**Table 2 toxins-13-00905-t002:** Studies reporting bioactivities originating from unknown bioactive compounds within the genus *Alexandrium*. Details about possible misidentification of strains are given in the footnote. No published information on bioactive compounds were found for the following species: *A. balechii*, *A. compressum*, *A. camurascutulum*, *A. concavum*, *A. depressum*, *A. foedum*, *A. fraterculus*, *A. gaarderae*, *A. globosum*, *A. kutnerae*, *A. satoanum*, *A. tropicale*, *A. acatenella.* Absence of references indicates that no information is available from the literature.

Species	Hemolytic	Anti-Pathogen	Allelopathy	Anti-Grazer	Toxicity to Bivalves	Ichthyotoxic (Fishes)	Cytotoxic
*A. affine*			[4]		[169]	[170]	
*A. andersonii*	[171]		[21]				[172]
*A. catenella*	[173] ^a^		[60] ^b^	[164] ^c^	[66]	[73]	
*A. fraterculus*			[21]				
*A. insuetum*			[121]				
*A. leei*						[174]	
*A. margalefii*			[121]				
*A. mediterraneum*			[121]				
*A. minutum*	[157]	[64]	[157]	[175] ^d^	[68]		[176] ^e^
*A. monilatum*	[143] ^f^					[177] ^f^	
*A. ostenfeldii*			[163]	[85]			
*A. pacificum*		[178] ^g^	[121]				
*A. pohangense*			[61]				
*A. pseudogonyaulax*			[62]	[65] ^h^			
*A. tamarense*	[173] ^i^		[72]		[66]		
*A. tamutum*			[153]	[179] ^j^			
*A. taylorii*	[180] ^k^		[3]	[180] ^l^			
Undefined strains of *catenella/tamarense/* *fundyense species complex*	[181] ^m^, [157] ^n^						[182] ^o^, [183] ^p^

(a) Strain of *A. catenella*. Hemolytic activity of cell extract but not of the filtrate. Ref. [48] previously identified as a *A. fundyense*. (b) Strain of *A. catenella* reported as *A. fundyense*. (c) Strain of *A. tamarense* reported as *A. fundyense*. (d) *A. minutum* reported as *A. lusitanicum*. (e) The role of PSTs cannot be excluded as the toxin profile of the strain is not given. (f) *A. monilatum* reported as *Gonyaulax monilatum*. (g) *A. pacificum* reported as *A. catenella*. (h) The strain of *A. pseudogonyaulax* used in this study contained Goniodimin A, the role of which in anti-grazing activity remains to be investigated. (i) Hemolytic activity of cell extract but not of the filtrate. (j) Strain of *A. tamutum* reported as *A. tamarense*. (k) No details (e.g., no strain identifier, no sequence data, and no morphology). (l) No details (e.g., no strain identifier, no sequence data, and no morphology). (m) Strain identified as *A. catenella* but no details (e.g., no strain identifier, no sequence data, and no morphology). Hemolytic activity of cells but not of filtrate. (n) Strain identified as *A. tamarense* but unlikely to be a true *A. tamarense* because of PSTs production. No sequence data and morphology found for this strain. (o) Strain identified as *A. tamarense* but no details (e.g., no strain identifier, no sequence data, and no morphology). (p) Strain identified as *A. catenella* but no details (e.g., no strain identifier, no sequence data, and no morphology).

### 3.1. Effect of BECs on Marine Organisms

When describing chemical interactions between photosynthetic plankton species and other marine organisms, the concept of allelopathy is often used. Allelopathy is commonly referred to in terrestrial systems and refers to “the impact of plants upon neighboring plants and/or their associated microflora and/or macrofauna by the production of allelochemicals; often these allelochemicals interfere with plant growth but they may also result in stimulation of growth” [184]. Applying the terrestrial concept of allelopathy on plankton systems, some authors only include phytoplankton–phytoplankton interactions [185,186,187]. However, in the marine realm of protist communities, the typical plant–animal dichotomy is not applicable: in fact, many planktonic organisms are mixotrophs, meaning that they combine photo- and heterotrophy [188,189]. Consequently, other authors broaden the allelochemical concept to comprise interactions of protists and bacteria in general, thereby including, grazers or any organisms competing with protists [4,162,163,190,191]. In this review, we follow such broad concepts defining allelochemical interactions as negative effects of a protist on accompanying organisms, through the release of chemicals, named allelochemicals. These BECs-based chemical interactions might cause various effects and thus may have several ecological roles. In the following chapters the effects are subdivided according to the target organisms and cells, including physiological incapacitation and toxicity to shellfish and fishes.

#### 3.1.1. Allelochemical Activities

##### Anti-Pathogen Activities

Species of *Alexandrium*, like other dinoflagellates, are subject to pathogens (e.g., viruses, parasites, and bacteria) that at times contribute to the loss of biomass [192] and even bloom demise. Exudates with allelochemical properties of *A. minutum* showed anti-parasitic activity against the Syndinialean Amoebophrya sp. in co-cultures [64]. Exposure of free-living infectious spores of Amoebophrya sp. to *A. minutum* filtrates rapidly resulted in a permeabilization of their membranes. This eventually led to a loss of virulence of the parasite in cultures. When the herpesvirus OSHV-1 µVar, a common pathogen of oysters *Magallana gigas* (previously referred as *Crassostrea gigas*) was exposed to *A. pacificum* (initially reported as *A. catenella*), the prevalence of viral infection in *M. gigas* decreased [178]. Potential direct deleterious effects of *Alexandrium* BECs on marine viruses may thus mitigate infection by decreasing the viral load. Effects of BECs on pathogenic bacteria have not been studied yet. Generally, the bacterial consortium associated with *Alexandrium* cultures are not negatively affected [162,193], rather their growth is enhanced by the presence of allelochemicals, and this increased growth might be due to the release of organic matter from lysing cells.

##### Effects on Protist Competitors

Allelochemical activity of *Alexandrium* strains on protist competitors is usually assessed using co-cultures, or by adding cell-free medium (either supernatant or filtrate) to test target organism responses [4,58,163,165,194,195]. In such experiments, it is usually reported that activity of whole cultures is slightly stronger than supernatant or filtrate [165], likely because in co-culture allelochemicals are continuously released during the incubation period. Nothing is known about potential modulation of allelochemical production/release in presence of target cells, either caused by cell-to-cell contact or by target-based chemical cues. The latter process is known for PSTs, where the production increases in response to chemical cues from dead microalgal cells that are potential competitors [196] and in response to copepod chemical cues [197]. Conversely, lytic potency can be significantly enhanced in *Alexandrium* under certain environmental conditions such as P-starvation [166], low light [167], reduced salinity [198], or upon Cu addition [199]. Generally, reduced growth or exhaustive growth cessation at stationary phase may lead to significant BECs accumulation [157].

Allelochemical activity of *Alexandrium* spp. is mostly reported as growth inhibition [60,195,200] or cell lysis [4,58,163]. Other visible effects on target cells in contact with allelochemicals of *Alexandrium* spp., such as aberrant behaviour (e.g., backward swimming [201]), immobilisation of motile cells [4], cell deformation [163], or cell bleaching [165,200] might be considered as pre-lytic stages. For some target species, the formation of temporary cysts in response to *Alexandrium* allelochemicals has been observed [162,163]. While the gross effects of BECs have been widely described, the precise mode(s) of action of allelochemicals remain(s) to be confirmed. The thorough studies of allelochemical interactions between *A. catenella*—*Rhodomonas salina* (Wislouch) D.R.A. Hill and R. Wetherbee [58,194,202], *A. minutum*—*Chaetoceros muelleri* Lemmermann [165,199,203,204], or co-cultures between *A. pacificum* (reported as as *A. tamarense*)—*Pheodactylum tricornutum* Bohlin [195] foster the hypothesis that allelochemicals of *A. minutum* and of species of the former *A. tamarense* species complex are fast-acting membrane disruptive compounds [202,204]. A membrane-disrupting effect was confirmed on artificial liposomes [202]. When testing photosynthetic target species, the fluorescence yield (estimated by pulse-amplitude modulation fluorometry) during the first 3 h of incubation, as long as cells were not lysed, did not significantly change, suggesting that allelochemicals produced by *A. ostenfeldii* cause no short-term negative effects on the photosynthetic apparatus [163]. In the long run, however, photosynthesis is also impaired with reports of blurred and disintegrated chloroplast membranes in *P. tricornutum* in co-culture with “*A. tamarense*” (not further characterised) [195] and inhibition of photosynthetic electron flow between the two photosystems [204]. The inhibition of photosynthesis seems to originate from an impaired electron flux that leads to larger inhibitions of photosystem I (PSI) and photosystem II (PSII) [165,203,204]. Eventually, the deleterious effects lead to the death and the lysis of microalgal cells.

Inhibiting, impairing, or even eliminating competitors provides an advantage with respect to competition for limiting resources. Another additional level of benefit is when organic matter of competitors (either particulate or dissolved) is used by the allelochemical donor species, a “kill and eat your competitor” strategy [205,206]. In fact, a number of *Alexandrium* species are known to be mixotrophs and to ingest other protistan species (Table 1, [22]), and here phagotrophy is likely to be supported by a chemically mediated prey immobilisation. Among phagotrophic species of *Alexandrium*, *A. pohangense* was shown to ingest *Margalefidinium polykrikoides* (Margalef) F. Gomez, Richlen and D.M.Anderson cells by engulfment, after immobilizing the prey cell using exuded materials [61]. Cells of *Alexandrium pseudogonyaulax* use a mucus trap for feeding. In co-culture with *A. pseudogonyaulax*, various protists are immobilised [62,120], and video observations of prey being immobilised in the mucus indicate that *A. pseudogonyaulax* might “fish” prey with a toxic net made of mucus and allelochemicals.

Lytic potency of mixotrophic *Alexandrium* might be partly responsible for an overestimation of grazing rates when their calculation is based on the decline of prey concentration (e.g., [61,157]) and when an unknown portion of prey cells were lysed and not ingested.

Allelochemicals produced by the genus *Alexandrium* affect a large range of marine protists including phototrophic and heterotrophic organisms, but not all species or strains are affected with the same intensity [58,59,162,163]. Moreover, pairwise mixing different targets with different *Alexandrium* (*A. catenella*, *A. ostenfeldii*, *A. minutum*) revealed different sensitivities of target species to allelochemicals of different *Alexandrium* species. This indicate that different *Alexandrium* species may produce different compounds or different “cocktails” of similar compounds [58]. When natural communities were exposed to a lytic *A. catenella* filtrate in controlled conditions [60,193] (reported as *A. fundyense*, and *A. tamarense*, respectively), a decrease in populations of ciliates and various other plankton organisms (e.g., diatoms, nanoflagellates) was observed except for the group of small (<30 µm) dinoflagellates. Field observations confirmed the same patterns: decrease in nanoflagellate and diatom abundances and increase in dinoflagellate densities [60] suggesting that dinoflagellates might have protective mechanisms. However, controlled co-culture experiments of *A. catenella* (reported as *A. tamarense*) and different target species revealed that growth of all five tested dinophyte species were negatively affected and four of the five species finally died off at the end [59]. Generally, self-protection of cells that produce bioactive extracellular compounds is an obvious need. As for allelochemicals of *Karlodinium veneficum* (detailed in the chapter “3.3.3 BECs of other microalgal species”), the protection might originate from the biochemistry of the membrane [202].

Another protective mechanism of some dinoflagellates against allelochemicals is the formation of temporary cysts [59,162], where the dinophyte cells shed their theca and stay alive in pellicle cysts and are able to hatch and grow once the allelochemical concentration decreased significantly. Cell size can also influence the sensitivity to allelochemicals [200], in the diatom *Thalassiosira* cf. *gravida*, the larger cells that had undergone sexual reproduction were more resistant to “*A. fundyense*” (not further characterised).

##### Anti-Grazing Activities

Allelochemicals produced by *Alexandrium* do not only affect protistan competitors but also protistan grazers. An “anti-grazing effect” is obvious when phagotrophic protists show aberrant swimming behaviour and are immobilised or lysed in presence of the putative prey or its BECs [4,85,163,164,201,207,208]. A case-study is the heterotrophic dinoflagellate *Polykrikos kofoidii* Chatton, an abundant and voracious grazer of large dinoflagellates such as *Alexandrium*. The feeding and growth responses of *P. kofoidii* to various *Alexandrium* spp. strains (including both PSP and non-PSP strains) varied depending on the strain. Some *Alexandrium* strains supported rapid growth of the predator, whereas others rapidly caused cell death of the heterotrophic dinoflagellate [209,210]. *Polykrikos kofoidii* was shown to be negatively affected by lytic *A. catenella* strains (reported as *A. tamarense*) while a non-lytic *A. catenella* strain Alex5 (reported as *A. tamarense* or *A. fundyense*, respectively) was ingested and allowed rapid growth of the grazer [165,211]. Deleterious effects of *Alexandrium* on *P. kofoidii* (and to the ciliate *Tiarina fusus* Bergh, 1981) were not related to PSTs [207]. Notably, in multi-strain incubations the presence of lytic compounds did not only protect the producer strains but also mitigated the grazing on a non-lytic strain, therefore protecting the whole community [164]. Negative effects of *A. catenella* (reported as *A. tamarense*) and *A. ostenfeldii* on the tintinnid *Favella ehrenbergii* (Claparède and Lachmann, 1858) Jörgensen, 1924 were initially speculated to be caused by PSTs [85,201], but in light of the data presented above, it is very likely that lytic compounds (not known at that time) and not PSTs in fact were responsible.

There is a plethora of evidence that *Alexandrium* spp. chemically affects non-protistan grazers of the metazooplankton [65,179,180,212,213]. While available data suggest that protists are not affected by PSTs, the neurotoxins have demonstrated anti-grazing effects against non-protist grazers [81,82]. An anti-grazing activity of a PSTs standard mixture was shown on the marine copepod *Tigriopus californicus* [81], while toxic effects of purified gonyautoxin (analogues of STX) standards were evidenced on the crustacean *Artemia salina* [82]. Another strong argument in favour of PST anti-grazing activity is drastic enhancement of PSTs production in the presence of some copepods or their chemical cues [197,214,215,216].

While many studies highlighted toxic effects of PST-producing cells on copepods (e.g., [213,217,218]), it must be noted that in most of the copepod—*Alexandrium* grazing studies BECs were not considered. However, few studies highlighted deleterious effects of non-PST but lytic strains. For example, lytic compounds of “*A. taylorii*” (*A. taylorii* is regarded as a non-PST-producing species [3], but note that in the study of Emura and collaborators [180] no strain identifier, no morphology and no sequence data were provided, which would allow confirmation of the *Alexandrium* species determination) were shown to be toxic against *Artemia salina* [180]. The crustaceans were immobilised and died when exposed to the supernatant or the whole culture. Non-neurotoxic compounds of *A. tamutum* (reported as *A. tamarense*) were also involved in reprotoxic effects against the copepod *Temora stylifera* [179].

The situation might be even more complex as anti-grazing activities may not be limited to PSTs or lytic compounds alone. In a comparative approach using lytic and non-lytic *Alexandrium* strains (based on a microalgal bioassay), as well as PST-producing and non-producing *Alexandrium* strains, Xu and collaborators [65] highlighted that the behaviour of the copepod *Temora longicornis* was modified (i.e., reduction of feeding activity or regurgitation of cells) when fed on *Alexandrium* spp., but the modification was neither linked to PST nor to lytic compounds. It seems that in fact several different groups of bioactive compounds are potentially involved in anti-grazing activities. A high diversity and potentially a target specificity of anti-grazing compounds might reflect different stages of the ever-going arms race between phytoplankton and grazers [219].

Overall, the anti-grazing compounds highlight the general problem of the simultaneous presence of different compound classes in *Alexandrium* and the associated problem of identifying the compounds responsible for a specific bioactivity. Since the compounds are not chemically identified it is impossible to attribute the different effects to a specific group of toxins. Further comparative analysis with *Alexandrium* strains with specific bioactivities (e.g., only PSTs, only lytic, only anti-grazing) or mixed bioactivities are required to assign observed effects to a single group of compounds or to a mixture of known toxins and BECs. This would allow to retrospectively evaluate the effects already observed on copepods [220] or benthic gastropods [175,221], for example.

#### 3.1.2. Toxicity towards Shellfish

Toxicity of *Alexandrium* spp. towards shellfish has been widely reported and historically associated to PSTs. Until recently, in most of such studies, only the PSTs content of the *Alexandrium* strain(s) was reported. A potential production of BECs was not assessed, which renders it impossible to clearly identify and distinguish the effects of PSTs from BECs. Several studies on shellfish hypothesised a production of BECs by experimental *Alexandrium* strains, but without actually assessing the hemolytic or allelochemical potency. This is for example the case of a study by Ford and collaborators [222], showing inhibition of adherence and phagocytosis of hemocytes from the clams *Ruditapes philippinarum* and *Mya arenaria* by a filtrate of a non-PST producing strain of *A. tamarense*.

A number of recent studies used a comparative approach with strains that were thoroughly characterised with respect to different bioactive compounds, enabling the distinction between effects from PSTs or BECs (Figure 2). In particular, some of these new studies [66,67,68,70,71,72] exposed bivalve species (*M. gigas*, *Mytilus edulis*, *Pecten maximus*) to different strains of *Alexandrium* spp., for which PST production and BECs potency (measured as allelochemical or hemolytic potencies) were characterised. These studies clearly highlight that a majority of the deleterious effects observed on bivalves might indeed be due to BECs produced by *Alexandrium* spp. rather than by PSTs. These observations might also explain the diversity and/or controversy of effects on fish and shellfish reported in the literature. Borcier and collaborators [67] highlighted harmful effects of extracellular compounds when juveniles of the great scallop *P. maximus* were fed with a mix of microalgae containing strain CCMI1002 of *A. minutum*, which only produces allelochemicals but no PST. Daily shell growth was delayed, tissues in direct contact with *A. minutum* cells were altered, and reaction time decreased when the scallops were exposed to predators. Similarly, different effects were visible when *M. gigas* was exposed to different strains of *A. minutum* producing either only BECs (strain CCMI1002), only PSTs (strain DA1257), or both (strain AM89BM) [68]. This later study demonstrated that the majority of the physiological effects (e.g., modification of valve-activity, hemocyte mobilisation in the gills) of *A. minutum* on oysters were attributable to BECs. Other comparative studies were conducted on mussels exposed to three different strains of *Alexandrium* [66]. Two strains of *A. catenella* (Alex2 and Alex5) were producing PSTs, but only one of them (Alex2) was also lytic. The third strain (*A. tamarense* NX-57-08) was lytic but non-PST producer. The authors demonstrated cytotoxic effects of bioactive (i.e., lytic) extracellular compounds. Effects of these lytic compounds comprised also other deleterious effects such as immune impairment or lysosomal membrane destabilisation. These immunosuppressive effects were further confirmed in vitro, on hemocytes from the mussel *M. edulis* exposed to a lytic supernatant of *A. tamarense* (strain NX-57-08) [72].

Early planktonic and larval stages of shellfish are also strongly affected by BEC. The *A. minutum* strain CCMI1002 (BECs but non-PST producer) showed deleterious effects of BECs on developmental stages of oysters. Specifically, this *A. minutum* strain caused lysis of embryos (the most sensitive stage to *A. minutum* toxicity among planktonic life-stages of oysters), inhibited hatching into D-larvae, disrupted larval swimming behaviour, feeding and growth, and induced decreases in survival and settlement of umbonate and eyed larvae [70]. Lytic activity of non-PST *A. affine* was reported on gametes, embryos and larval stages of the pearl oyster *Pinctada imbricata* (reported as *Pinctada fucata* martensi) [169,223]. The deleterious effects of “*A. catenella*” (not further characterised) exudates were also evidenced upon Pacific oyster (*M. gigas*) and Mediterranean mussel (*Mytilus galloprovincialis*) larvae [224]. The deleterious effects comprised larval mortality and histopathological changes of the digestive apparatus. The lytic potency of the particular *Alexandrium* strain was not assessed but it is likely that BECs are involved in these effects.

Now that some specific *Alexandrium* strains have been better characterised with respect to their PSTs and BECs production potential, it is instructive to retrospectively evaluate earlier exposure studies using these strains. This is especially the case with the *A. minutum* strain AM89BM (lytic and PSTs), which was used in many previous exposure experiments [54,225,226,227]. The immuno-suppressive effects on hemocytes from the eastern oyster *Crassostrea virginica* [54], or the production of reactive oxygen species by *M. gigas* gametes [227] exposed to the supernatant of this *A. minutum* strain might retrospectively be attributed to lytic compounds. A modelling approach (Dynamic Energy Budget modelling [228]), using the AM89BM strain, also revealed putative effects of *A. minutum* BECs on the physiology of oysters. Knowledge of the production of lytic compounds by this strain [67,68], enabled the authors to better constrain their model when they considered the production of BECs. They hypothesised that BECs alter the feeding behaviour of oysters, causing a decrease in the ingestion rate.

#### 3.1.3. Ichthyotoxicity

In addition to the numerous studies regarding PSTs accumulation and/or direct effects of *Alexandrium* on shellfish, there are a number of cases, where blooms of *Alexandrium* spp. were linked to fish kills [56,57,229,230,231,232,233,234]. Toxicity of unknown compounds toward fish was first shown with *A. monilatum*, a species not identified as a PSTs producer (Table 1), on the guppy *Lebistes reticulatus* [177]. Signs of hypoxia were observed on the surf smelt *Hypomesus pretiosus* japonicus exposed to culture filtrate of two *Alexandrium* strains (reported as “*Protogonyaulax catenella*” and “*P. tamarense*”, not further characterised) [235]. The ichthyotoxicity was attributed to extracellular compounds as the filtrate was free of cells and PSTs (estimated with a mouse assay). Extracellular toxins were found to swell and exfoliate fish gills, as well as decreasing membrane elasticity of rainbow trout eggs. Similarly, the ichthyotoxicity of *A. leei* against Asian sea bass fishes was not correlated to PSTs but was attributed to extracellular compounds present in cell-free culture medium [174]. Toxicity of supernatants of different strains of *A. catenella* was also observed on cell lines of rainbow trout (RTgill-W1) [73]. The toxicity varied with strains and sample source (i.e., whole culture, cell extracts, or supernatant) but was not correlated to the varying PSTs content of the different strains, and was therefore linked to other group(s) of compounds. In a later study [170], the exposure of red seabream to cells of non-PST *A. affine* resulted in the death of fishes. Sublethal effects of *A. affine* on fishes comprised decrease in respiration rates, immunosuppression and hepathic impairment in tissues. Although the authors concluded that deleterious effects originated from the attachment of the dinoflagellate to gill tissue, they did not take into consideration the potential production of unknown BECs and its role in ichthyotoxicity. Regarding the putative ichthyotoxicity of BECs, in the absence of their chemical identification, it is essential to compare the effects of only-BECs and only-PST *Alexandrium* strains on fish gill cells and whole fish to further evaluate their role in fish kills.

### 3.2. Methods for the Detection, Quantification and Study of BECs

Standard methods for the detection and quantification of phycotoxins involve chemical methods such as high-performance liquid chromatography coupled to mass spectrometry (HPLC-MS) or to a fluorescence detector (HPLC-FLD), or immunosorbent assays (i.e., enzyme-linked immunosorbent assay; ELISA). BECs, however, are largely uncharacterised compounds and are thus impossible to detect and quantify using these chemical methods. Nevertheless, several bioassays based on different bioactivities (e.g., lysis of blood cells or protists, effects on microalgal photosynthesis, or ichthyotoxicity) have been used to detect and quantify BECs in culture or cell extracts. As opposed to classic experiments made to characterise the toxic effects, bioassays can be defined as biological tests where organisms living under laboratory conditions are exposed to various concentrations of a toxicant, thus allowing quantification of their potency. In the future, a limited number of standard bioassays should be favoured to study BECs thus allowing better comparison between studies and between algal species and strains. In the case of bioassays with cell lines (e.g., hemocytes, microalgae, and fish gill cell lines), target strains available from culture collections should be favoured.

There are some methodological aspects and challenges when working with BECs, and not considering these in the experimental protocols might lead to substantial losses of BECs, and consequently, to misinterpretations. The first crucial step is to decide which fraction of a culture or field sample to analyse. Bioassays for the study of BECs can be performed with supernatants, filtrates or cell extracts of microalgal cultures or field samples. Some filters can retain allelochemical activity and therefore the use of supernatants [194] or specific filters (acetate cellulose or asymmetric polyethersulfone) is recommended [156]. Care must be taken during the storage of samples: it is recommended to limit the use of plastics [194] that are capable of adsorbing some allelochemical BECs. If the use of plastics cannot be avoided, their utilisation should be mentioned. Available data on BECs indicate that they are relatively resistant to cold, heat, and pH changes [73,165,174,182,194].

#### 3.2.1. Hemolytic Bioassay

It is unlikely that blood cells are directly exposed to the toxins and thus are not the ecologically relevant targets, but blood cells of various organisms (including humans) are commonly used in toxicity assays. In haemolytic bioassays, red blood cells are exposed to samples and the lysis of erythrocytes (haemolysis) is quantified fluorometrically. Haemolytic activity of compounds in the culture medium have been shown for different *Alexandrium* species (Table 2 and [157,173,180,182,235,236,237]). Haemolytic activity of *Alexandrium* spp. is not correlated to PSTs content [236,237] but is positively correlated with *Alexandrium* spp. cell density and culture age [157]. Haemolytic activity is usually accompanied by other bioactivities such as allelochemical activity [157,236,237], and anti-grazers activity [180]. While haemolytic bioassays are reproducible and efficient in highlighting extracellular toxicity, these bioassays are not specific to give information on the ecological roles and potential targets of the compounds: haemolytic and allelochemical potency did not always correlate [236], suggesting the production of several BECs.

#### 3.2.2. Protistan Bioassay

Allelochemical potency of *Alexandrium* spp. is most often quantified through inhibition of growth or photosynthesis and through lysis of other protists. *Alexandrium* affects many marine protists including photototrophic and heterotrophic organisms [58,59,162,163] so there is a wide range of bioassay target species available. However, variability in the sensitivity of microalgal species and strains to *Alexandrium* spp. allelochemicals [58,59,238] has to be considered, and using exclusively a resistant target strain would lead to false conclusions. For the better comparability of BECs detection and quantification between different studies and laboratories, we therefore recommend the use of a limited number of strains that are available in culture collections.

A frequently used protistan bioassay is the *Rhodomonas salina* assay [58,66,161,163,166,167,194,198,239,240]. In this bioassay, the cryptophyte *R. salina* (e.g., strain KAC 30 from the Kalmar culture collection) is exposed to donor species whole cells, culture filtrates/supernatants or cell extracts, for a defined incubation period (usually 3–24 h). The decrease in cell density compared to a control is then quantified microscopically (e.g., [163]), but can also be quantified fluorometrically [241]. This bioassay has also shown its benefits in bioassay-guided fractionation protocols to purify allelochemicals [194,239]. Another bioassay [156] used in several studies [67,68], is based on the inhibition of photosynthesis in the diatom *Chaetoceros muelleri* (Strain CCAP 1010/3 from the culture collection of algae and protozoa (CCAP)). In this bioassay, *C. muelleri* cells are exposed to *Alexandrium* spp. filtrates or supernatants for 2 h before quantifying the inhibition of maximum photosystem II quantum yield (Fv/Fm), which has been identified as a sublethal effect before cell lysis [204]. These two bioassays are based on different target species and therefore it is possible, that in fact, different compounds are quantified, i.e., compounds inducing cryptophyte lysis versus compounds inducing diatom lysis. To clarify this issue, extended intercomparisons of both bioassays are needed.

#### 3.2.3. Anti-Grazer Bioassay

Next to classical grazing experiments [242,243] there seem to be no standardised bioassay-type setups to specifically study anti-grazing bioactivities of *Alexandrium*. A widely applied bioassay uses the brine shrimp Artemia spp., which is at least a potential grazer of microalgae (e.g., [244,245]). However, this bioassay does not specifically quantify anti-grazing or repellent activity. Artemia spp. individuals are exposed to microalgal cells or filtrates, then immobile or dead Artemia individuals are quantified to estimate lethal concentrations. In the context of *Alexandrium*, the *Artemia* assay has been used to highlight the presence of “*A. taylorii*” (not further characterised) exotoxins [180]. This bioassay highlighting toxicity should be complemented with a more specific bioassay quantifying ingestion rate of *Alexandrium* cells by zooplankton (e.g., [246]) to highlight anti-grazing activity. Video observation and quantification of ingestion or rejection events of *Alexandrium* cells by copepods, as performed in [65] might also be adapted in a bioassay to quantify anti-grazing bioactivities.

#### 3.2.4. Ichthyotoxicity Bioassays

In the first ichthyotoxicity bioassay highlighting *Alexandrium* BECs [174], whole fishes (*Later calcacifer*) were exposed to *A. leei* cultures, filtrates or different extracts for 3–7 days. The percentage of mortality, the death time, and the size of fish were then estimated. This bioassay highlighted that ichthyotoxicity of culture originated from exudates and allowed a partial characterisation of the chemical properties of *A. leei* ichthyotoxin(s). While whole fish bioassays give robust and ecologically relevant results, they require special aquarium facilities and ethics approvals to work with whole fish. Later, the use of ichthyotoxicity bioassays using rainbow fish gill cell lines [247,248,249] was applied to the genus *Alexandrium* [73,250]. In a first approach, Mardones and collaborators [73] monitored gill cell (RTgill-W1) lysis with a fluorometric assay, later the Ca^2+^, H^+^, and K^+^ ion fluxes were measured with microelectrodes [250], following the addition of culture supernatant or cell extract. Both versions of the fish-gill bioassay provide rapid results after 1–2 h of incubation and allow the detection and quantification of ichthyotoxic BECs in cultures. Thus, in vitro fish-gill bioassays allow a rapid screening of samples and, when performed under controlled conditions, allow comparative studies [248] between different *Alexandrium* species or strains. However, it has to be kept in mind that the response of isolated gill cells does not necessarily represent the complex response of the whole organism. No specific bioassay has been developed for shellfish but bivalve haemocytes and gametes isolated from whole organisms were used in many studies [72,222,227,251] and can allow a rapid detection (2 h) of BECs toxicity to bivalves [227]. However, there are currently no long-term marine invertebrate cell lines available. The use of standardised and free of pathogen shellfish [252] such as the standardised oyster seed used in [253] should be generally considered and might improve the comparability between studies and between different laboratories.

### 3.3. State of the Art of Characterisation of Alexandrium BECs

Attempts of a chemical characterisation of bioactive extracellular compounds produced by *Alexandrium* spp. have been based on the different bioassays and thus potentially on different bioactivities. The standard haemolytic assays are described in Section 3.2.1. Haemolytic bioassays have been used for some early attempts to characterise BECs of *Alexandrium* [180,182]. Other studies focused on the characterisation of BECs based on their allelochemical potential [183,194,195,207,239,254] or on ichthyotoxic effects [73,174,255]. This section compiles the information regarding various attempts to characterise BECs, including detailed information on the various sample preparation or extraction protocols (Table 3) and an overview of current knowledge and hypotheses about the nature of BECs (Table 4).

#### 3.3.1. Characterisation of BECs Based on Haemolytic and Allelochemical Properties

Different hypotheses were provided regarding the nature of *Alexandrium* allelochemicals and were controversial (Table 4), especially regarding the potential proteinaceous nature. Although it is not proof of their implication in bioactivity, the presence of polypeptides was reported in the supernatant extract of an allelochemical strain of “*A. tamarense”* (not further characterised) by ultra-performance liquid matrix-assisted laser desorption/ionisation mass spectrometric (UPLC-MALDI-MS) [195]. In *A. minutum* cellular extracts with anti-cancer activity, three major proteins were found in the active fraction [176]; however, the activity of the extracellular fraction remains to be investigated. In these two studies, proteins were detected; however, their implication in activities was not thoroughly investigated (i.e., the activity of the purified polypeptides/proteins was not tested nor was a mitigation of the activity with a peptidase examined) and remains to be confirmed. A potential role of proteins in lytic activity was first hypothesised by Emura and collaborators [180]. The authors studied the haemolytic activity of a cell-free supernatant of “*A. taylorii*” (not further characterised) culture. They based the assumption of a proteinaceous compound on the temperature-dependant activity of the lytic activity but also on the mitigation of this activity when trypsin (a peptidase) was added. The authors further identified that lytic activity was associated with large compounds (>10 kDa) as the activity was concentrated by ultrafiltration and no activity was measured in the filtrate. Flores and collaborators [207] presented data related to the involvement of proteinaceous compounds in lytic activity of *Alexandrium* strains (all three reported as *A. tamarense* species complex; two strains likely to represent *A. catenella* as they originate from the US east coast and have PSTs, and one strain of *A. tamarense*) against the ciliate *Tiarina fusus* and the dinoflagellate *Polykrikos kofoidii*: the addition of peptidase (trypsin) mitigated the lytic activity of *A. tamarense* only, but toxicity of both *A. catenella* strains were not mitigated by trypsin addition. It is unclear at present, however, if these differences reflect assay or strain variability or true species–specific differences between *A. catenella* and *A. tamarense*. Neither hemolytic and cytotoxic activities from “*A. tamarense*” (not further characterised) exudates [182,236] nor the lytic fraction from *A. catenella* [194,239] revealed proteinaceous characteristics (trypsin or heat sensitivity, or Coomassie staining). Next to the potential proteinaceous nature of *Alexandrium* BECs, Flores and collaborators [207] also suggested a possible involvement of reactive oxygen species as they observed an alleviation of lytic activities after the addition of antioxidant enzymes (peroxidase and superoxide dismutase). Contrasting results on the proteinaceous nature of BECs might be explained by different *Alexandrium* species releasing different groups of lytic compounds with different nature, e.g., non-proteinaceous compounds in “*A. tamarense*” (not further characterised) [182,236] and protein-like compounds in “*A. taylorii*” (not further characterised) [180] and in *A. tamarense* [207]. However, the unclear species identification in some studies [180,236] precludes such interpretations.

Another candidate compound was further isolated by electrophoresis from “*A. tamarense*” (not further characterised) cell-free medium [182]. The putative “AT-toxin” has a high molecular weight, estimated to be around 1000 kDa, and does not contain significant amounts of amino acids but contains sugars. The sugar composition was thoroughly described by the authors who consequently hypothesised that an extracellular polysaccharide-based toxin was responsible for haemolytic activity and induction of apoptosis in a human myeloid leukemia cell line. Similarly, the presence of saccharides has been reported in an *A. minutum* cytotoxic fraction with anti-proliferative activity against cancer [176]. In the later study, the authors identified that the bioactive fraction contained glycoprotein(s) with molecular weight(s) above 20 kDa, the involvement of which in activity remains to be investigated. An involvement of sugars in bioactivities remains to be confirmed through further purification steps and/or by testing the bioactivity of the extract following polysaccharide degradation. In contrast, in *A. catenella* (reported as *A. tamarense*) the role of sugar in lytic activity from culture supernatant seems unlikely [239].

Hydrophobicity is a shared feature of BECs from at least three species, *A. catenella* (reported as *A. tamarense*) [194,202,239], *A. tamutum* (reported as *A. tamarense*) [179] and *A. minutum* [254]. Characterisation of hydrophobic *Alexandrium* BECs has mainly relied on methanolic extraction of allelochemical activity from the supernatant [194,239,254]. Allelochemicals were extracted from *A. catenella* and *A. minutum* supernatant through solid phase extraction (LC-18 SPE) allowing separation of a hydrophobic fraction (eluted with 80% of methanol) which was much more potent than the others [194,239,254]. A common notion, which is in accordance with the hydrophobic nature, is that during preparation and sample handling there are significant losses of biological activities probably due to stickiness to surfaces, with plastic material causing larger loss compared to glass. When stored in glass flasks, an apparent “loss” of activity was observed but could be recovered by vigorous shaking [194]. Allelochemicals of *A. catenella* (reported as *A. tamarense*) were hypothesised to be amphipathic (i.e., compounds with a polar head and a hydrophobic part) according to liquid/liquid partitioning [194,239]. Based on further separation methods (ultrafiltration, reversed phase chromatography, and hydrophilic interaction ion-chromatography) followed by LC-TOFMS analysis, the authors eventually concluded that the putative allelochemicals of *A. catenella* are large (7–15 kDa), non-proteinaceous, and probably non-polysaccharidic molecules, but they could not isolate defined mass candidates. Such a large size (which is common in surfactants and detergents) corroborates the nature of allelochemicals as amphipathic compounds. Involvement of fatty acids as potential allelochemicals was much less likely, because the n-hexane phase of liquid/liquid partitioning did not exhibit allelochemical activity [194,239].

Despite the numerous attempts to characterise BECs of *Alexandrium* spp., only one molecular candidate from a single strain from “*A. catenella*” (not further characterised) was structurally identified: alexandrolide (Figure 1), a microalgal growth inhibitor [183]. Alexandrolide is a non-proteinaceous, non-saccharidic, hydrophobic compound isolated by bioassay-guided purification. The allelochemical with a molecular formula of C_28_H_49_NO_8_ and a pseudo molecular weight [M + H]^+^ of 528.3537 Da has a maximum ultraviolet absorption of 235 nm and infrared absorption at 1640 and 3400 cm^−1^. The fraction containing the newly identified molecule inhibited the growth of microalgae and was cytotoxic to mouse lymphoid P388. It was particularly potent against *Skeletonema costatum* (Greville) Cleve and *Chattonella antiqua* (Y.Hada) C.Ono but had much lesser effects on three dinoflagellate species tested. Notably, the chemical structure of alexandrolide is unlikely to induce a permeabilization of membranes as observed with some BECs. Experiments adding purified alexandrolides or determination of intra- and extracellular levels of alexandrolides are not yet available so its allelochemical function cannot be regarded as proven yet. Development of protocols is needed to allow the detection and quantification of alexandrolide in order to properly address its role in *Alexandrium* allelochemical interactions.

#### 3.3.2. Ichthyotoxic BECs

The study by Tang and collaborators [174] is the first to report basic chemical information on non-PST ichthyotoxins produced within the genus *Alexandrium*. The authors evidenced the presence of heat-stable and water-soluble ichthyotoxin(s) in exudates of a *A. leei* culture inducing the death of fishes (*Lates calcarifer*), but the precise nature of the ichthyotoxin(s) remains unknown. Later studies [73,255] investigated the nature of ichthyotoxins of *A. catenella.* These authors highlighted the presence of light sensitive, heat resistant and hydrophilic compounds [255] in both culture supernatant and extracts from lysed cells through an in vitro fish gill assay.

Mardones and collaborators [73] did not perform a bioassay-guided purification of ichthyotoxic extracts but tested the hypothesis that a synergistic action of reactive oxygen species (ROS) and free fatty acids (FA) were responsible for ichthyotoxicity, as was proposed for *Chattonella marina* [256]. Therefore, they investigated the lipid composition of *A. catenella* strains and highlighted that polyunsaturated fatty acids (PUFA) accounted for more than 50% of total FA including docosahexaenoic acid (DHA; 22:6n-3), that counted for 16 to 20% of total FA [73]. Then, they tested the ichthyotoxicity of these FA alone or in combination with ROS in fish-gills based bioassays. DHA was highlighted to be the most ichthyotoxic FA to fish larvae and its toxicity was greatly enhanced when coupled to superoxide anions. It was thus hypothesised that oxidation products of FA play a dominant major role in ichthyotoxicity [73]. This toxicity would be particularly relevant for microalgal cells in contact with gills: the breakdown of cells would release FA and superoxide anions and result in lipid peroxidation and gill damages [256]. However, this scenario is unlikely to explain the ichthyotoxicity of supernatants that are free of cells, unless cells can exudate FA into the culture medium or centrifugation resulted in a significant breakdown of cells. The presence of significant amounts of FA or peroxides in culture supernatant has not yet been proven. It is thus still an open question if and how PUFAs and superoxide anions are involved in *Alexandrium* ichthyotoxicity.

#### 3.3.3. BECs of Other Microalgal Species

To better understand *Alexandrium* BECs activity, it is instructive to compile and compare knowledge about other toxins of microalgal origin known to be involved in the several toxic and lytic activities. Such compounds are karlotoxins produced by *Karlodinium veneficum* [257,258,259,260,261,262], karmitoxin produced by *Karlodinium armiger* Berholtz, Daugbjerg and Moestrup [263], amphidinols produced by *Amphidinium* spp. [264,265], or prymnesins produced by the haptophyte *Prymnesium parvum* N.Carter [266,267]. However, the allelochemical potency of prymnesins still remains to be confirmed [268]. All these ichthyotoxins, are polyketides, like the majority of dinoflagellate toxins [269]. Polyketides are a highly diverse class of secondary metabolites that are produced through polyketide synthases [270,271]. Polyketide diversity is based on variable compound structures (Figure 3A) and on the diversity of biological activities [272], including cytotoxic, ichthyotoxic, and hemolytic activities [273]. Notably, all the previously mentioned ichthyotoxins (i.e., karlotoxins, karmitoxins, prymnesins, and amphidinols), are putative allelochemicals and share a similar chemical feature: a hairpin structure with a polar head and a hydrophobic structure.

Karlotoxins, produced by *Karlodinium veneficum* [262] are particularly well described and reveal many analogies with BECs from *Alexandrium*. Karlotoxins are lytic compounds with ichthyotoxic activity [274], are toxic to bivalves [275,276], have anti-grazing activities [258,277,278], and have anti-parasitic activity against the genus *Amoebophrya* [279,280]. Karlotoxins are amphipathic compounds [281] with hairpin conformation including a large hydrophobic part that non-specifically increases membrane permeability [282]. Karlotoxins are suggested to form pores in membranes. The hydrophobic part of the toxin is inserted within the membrane and likely interacts with specific phospholipids, while the hydrophilic part is outside the membrane and interacts with cholesterols [283]. This organisation of toxins in membranes generates pores that result in membrane depolarisation, disruption of cellular functions [262], and eventually lead to anosmotic lysis of the cells. Karlotoxins, amphidinols, and karmitoxin are variants of the same core structure. While prymnesins are chemically quite different, they also share a common feature with the hairpin-like toxins: a large hydrophilic and a smaller lipophilic part. This amphipathic structure is likely linked to their affinity to cholesterols [265,284] and the disruption of membranes (Figure 3B). Polyketides are a promising track in the investigation of the chemical structure of the unknown bioactive compounds produced by the genus *Alexandrium*, regarding the variety of their bioactivities and mode of action.

**Figure 3 toxins-13-00905-f003:**
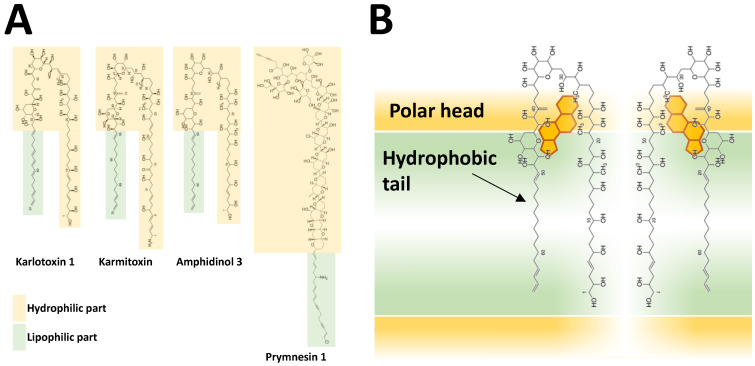
(**A**) Polyketides sharing the structures associated with the permeabilization of membranes and lytic activity. The green boxes represent the large hydrophobic part, and the yellow boxes represent the hydrophilic part of the toxin. (**B**) Proposed pore formation of the cell membrane by multiple karlotoxin molecules according to [283]. The simplified biphasic/lipidic membrane is represented by the rectangles with lipid polar heads in yellow and their hydrophobic tails in green. Karlotoxin are surrounding cholesterol molecules (in orange).

## 4. Synthesis

### 4.1. Ecological Role of BECs

Paralytic shellfish toxins (PSTs) are well known and thoroughly described in terms of chemistry and toxicology; however, their ecological role remains to be fully understood [285]. While PSTs are potent neurotoxins to mammals and seabirds, it is unlikely that their ecological function is to block mammalian ion channels [286]. Therefore, these toxins were hypothesised to act as pheromones [287] or as grazing deterrents [286], with the latter hypothesis substantiated by experimental evidence with copepods. Toxic effects to mammals are likely only collateral damages as no significant and direct biological interactions exist between them.

The ecological role(s) of BECs of the genus *Alexandrium* are also difficult to evaluate, especially when considering the wide range of organisms affected. The present literature review clearly emphasised the fact that many *Alexandrium* species and strains, even without producing PSTs or the known toxins, exhibit various bioactivities. While it is unknown whether all effects on marine organisms are based on the same or on a variety of different compounds, the different bioactivities investigated in a few particular strains suggest that they might be related (Table 5). Even though BECs might give a competitive advantage to *Alexandrium* by decreasing the grazing pressure of shellfish and or other benthic filter-feeders, *Alexandrium* are plankton organisms and thus toxicity to bivalves and ichthyotoxicity may be interpreted as collateral damage from the planktonic chemical warfare.

The haziness around the chemical identity of BECs and their putative variety make it impossible to conclude on a main role. Allelochemicals are usually discussed in association with the competition for resources or to facilitate feeding, but they can also be involved in defence against pathogens or predators [186,191]. Allelochemicals of *Alexandrium* might explain how *Alexandrium* blooms persist for longer periods and can reach extreme densities [6,9]. Allelochemicals provide a competitive advantage to *Alexandrium* cells by inhibiting competitors and parasites, and by limiting predation and grazing. Allelochemical interactions might also be a strategy developed by various mixotrophic species to immobilise prey [206,288,289] thus enhancing their competitiveness.

Understanding the evolution of allelochemical production and the variation of this genotypic trait in genotypically diverse dinoflagellate populations is challenging. The production of BECs might first be seen as a private good benefiting only the producer. But above a cell concentration threshold, the private BECs might become a “public” good [290] and favour the whole community by decreasing grazing, competition, and parasitism. This intraspecific facilitation has already been experimentally evidenced, where the presence of BEC-producing cells was protecting non-lytic cells from the same species against grazers [164] or cells from another genus against parasitism [64]. Both the rapid dilution of extracellular compounds and the related effective cell density threshold have been raised as a limit to the public good [291], as sufficiently high abundances would only be observed under bloom conditions and allelochemical are thus unlikely to support bloom development. However, at micro-scales where cell interactions occur, phytoplankton can form dense patches with concentrations orders of magnitude higher than background [292,293,294,295].

### 4.2. Toward the Essential Identification of BECs

To better understand compound-specific cause/effect relationships it is necessary to evaluate the presence/absence of both the known toxins and unknown BECs before assessing the effect of an *Alexandrium* strain on other organisms. As long as the BECs metabolites are not identified, the bioassays previously described can be used to quantify the different bioactivities. Even though some BECs bioactivities may be linked to a single class of compounds, one single bioassay probably will not yield a sufficient proxy for the quantification of the different bioactivities. Studies performing simultaneous analyses of different bioactivities of several strains with standard bioassay (e.g., [268] for *Prymnesium parvum*) are needed to better disentangle the possible link between activities.

To fully understand the ecological roles of BECs and their putative ichthyotoxicity and toxicity to bivalves, it is essential to isolate, characterise, and assess the diversity, the compound-specific toxicity, and mode of action of BECs. With respect to structural characterisation, a broad variety of different suggestions on the chemical nature of BECs can be found in different studies, such as proteinaceous compounds, saccharides, fatty acids, polyketides, and reactive oxygen species (ROS). To shed more light on this confusion, it still has to be worked out if, and to what extent, different species of *Alexandrium* may indeed produce different BECs. From literature data this is sometimes difficult to evaluate, as species determinations of the studied *Alexandrium* strains are not always sufficiently and reliably documented. Moreover, the chemical nature of *Alexandrium* BECs has so far only been investigated on semi-purified extract, and for only four out of the 17 *Alexandrium* species for which BECs have been reported. Next to *Alexandrium* species specificity, inconsistencies in BECs type suggestions likely originate also from varying experimental protocols for sample treatments and compound purifications. Partially, these protocols follow bioassay-guided fractionation, whereas others follow natural chemistry approaches without any functional control or verification of toxicity. Furthermore, a plethora of different treatments between working groups has been used in these studies that result in different biases in each of them. These different methods include precipitation of culture medium by high pH and solvation of the precipitate, ultrafiltration with different cut-offs, or dialysis. Also, loss of some BECs due to stickiness and surface adsorption has not been considered by many studies.

Compilation of available information emphasises that the chemical nature of BECs of *Alexandrium* remains largely unknown at present: only alexandrolide has been structurally elucidated so far [184]. It is now necessary to determine naturally occurring alexandrolide abundances and study the activity and mode of action of this compound at environmentally relevant concentrations. If these experiments indicate ichthyotoxic or allelochemical effects, it will become interesting to screen other *Alexandrium* species for this or related compounds. In any case, alexandrolide does not seem to share the chemical properties evidenced for *A. minutum* and *A. catenella* allelochemicals.

In the past, full identification of *A. minutum* or *A. catenella* BECs was hampered by several constraints: (i) low yields of extraction and purification, (ii) possibly large molecule size and amphipathic nature, and (iii) complexity of the exometabolome (i.e., chemical noise).

The substantial losses of allelochemical potency during the purification process does not ease their identification. The allelochemicals of the genus *Alexandrium* are particularly prone to adsorption on plastics [194,239]. Pre-filtration of fractions is one more step that induces a substantial loss of compounds. Optimisation of the protocol (e.g., investigating different filters, limiting the number of filtration steps or substituting filtration by centrifugation) is necessary to minimise losses. This implies that standard techniques of natural product chemistry may not be adequate for large amphipathic compounds. In addition, chemical noise is a major barrier for the identification of BEC. While comparative metabolomic (between lytic and non-lytic strains) is a way to identify candidates, allelochemicals still have to be isolated for further characterisation. As allelochemicals are likely to be potent at low concentrations, their signal is likely below the chemical noise (detection limit) of the fraction. While there is no evidence for a bacterial origin of allelochemicals [58,162,164], exudates of bacteria also add complexity to the organic fraction. Using axenic cultures [296] and artificial seawater [254] could ease the purification process by decreasing fraction complexity. Moreover, culturing conditions could be optimised to increase BECs yield. An increase in allelochemical potency was reported upon exposure of *A. minutum* to toxic concentrations of copper and allelochemicals were hypothesised to be copper chelators [199]. It may be possible to take advantage of this potential feature to isolate allelochemicals using immobilised metal affinity chromatography. This technique is used for the isolation of proteins exhibiting affinity to metals [297,298] and may be relevant to isolate *A. minutum* allelochemicals. Overall, new and standardised purification techniques based on the unusual behaviour of BECs should be applied to isolate them and ease their characterisation.

## 5. Concluding Remarks

The toxic effects of unknown/uncharacterised BECs of *Alexandrium* have largely been described on a wide range of marine organisms. These compounds can induce lysis of protists but also of haemocytes of various animals and mammalian cell lines. Regarding their various ecological functions/roles (affecting competitor, grazers, and parasites) BECs are likely to play an essential role in the dynamic of *Alexandrium* blooms. Despite numerous attempts to characterise them, the chemical nature(s) of BECs remain(s) poorly known or uncertain. The best-described BECs are those from *A. catenella* and *A. minutum* with allelochemical activity. They comprise unknown large amphipathic compounds that presumably target cell membranes and induce cell lysis. One compound, alexandrolide, a putative microalgal-growth inhibitor is structurally identified, but its ecological relevance remains to be confirmed. This review highlights the need to further characterise and quantify the chemicals responsible for numerous extracellular bioactivities. Due to BECs’ unusual chemical behaviours, new purification techniques are required to isolate BECs before assessing their characterisation. While BECs might be a limited group of compounds with wide bioactive spectra, the genus *Alexandrium* comprises several species, therefore potential chemical differences between species should be considered. It is essential to properly identify and document *Alexandrium* species, as this may be a source of inconsistencies between studies. BECs might have effects on the outcome and conclusions of studies targeting interactions of *Alexandrium* spp. and other marine organisms (especially ichthyotoxic and mixotrophy related studies). When performing such experiments, extracellular bioactivities should always be characterised together with detection and quantification of other known toxins. Until the nature of BECs is elucidated, we recommend the use of standard bioassays to quantify the different bioactivities. Intercalibration between bioassays should also be performed to shed light on the link between the different bioactivities. Once their structure(s) have been elucidated, the effects of purified compounds, and mixtures of compounds, on other cells should be screened and characterised, not only for toxicology purposes but also for potential biotechnological applications (e.g., anti-cancer [172,176], antiviral, antibacterial, and antiparasitic applications).

## Figures and Tables

**Figure 1 toxins-13-00905-f001:**
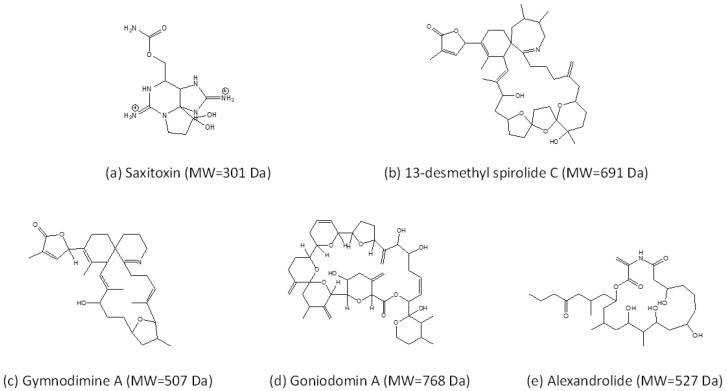
Examples of the diversity of toxins produced within the genus *Alexandrium*. (**a**) Saxitoxin, one of the most common PSTs; (**b**) 13-desmethyl spirolide C (SPX 1) and (**c**) Gymnodimine A (GYM A) are cycloimine toxins; (**d**) Goniodomin A (GDA) is a macrocyclic polyketide; (**e**) Alexandrolide is a putative allelochemical.

**Figure 2 toxins-13-00905-f002:**
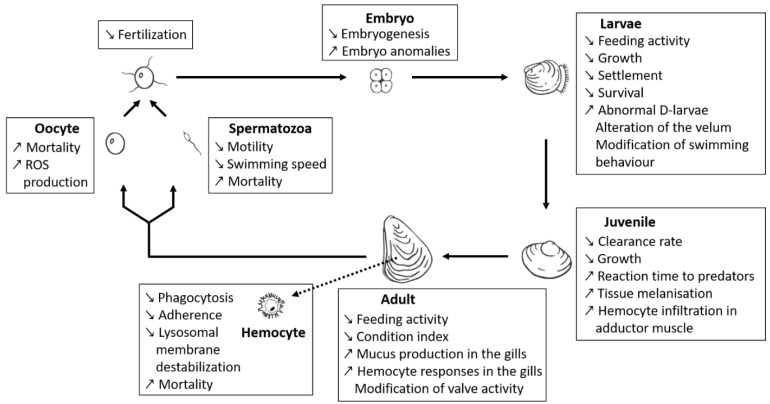
A general overview of the current knowledge from the literature on the effects of BECs produced by *Alexandrium* on different life stages of various marine bivalves (exemplified here for oysters). Bold arrows highlight the life cycle of marine bivalves, life stages are indicated in bold in the box. Small arrows in the boxes indicate significant differences that are either lower or higher from control: downward arrow indicate a decrease and upward arrow indicate an increase in the parameter as compared to the control.

**Table 3 toxins-13-00905-t003:** Diversity of protocols used for the chemical characterisation of *Alexandrium* BECs (and unknown intracellular cytotoxins).

Species	Bioactivity (Target Cell)	Sample	Method	Purification Steps						Chemical Analysis	Study
*A.* *andersonii*	Cytotoxic (Anti-cancer)	C	Bioassay guided purification	MeOH extraction	SPE Chromabond HR-X (CH_3_CN/H_2_O 7:3 and CH_2_Cl_2_/CH_3_OH 9:1)					Staining, NMR, LC-MS	[172]
*A. catenella*	Protist (*Rhodomonas salina*)	F	Bioassay guided purification	SPE C18 (80% MeOH)	Evaporation and redisolvation in water	HPLC C8 (18–19 min)	or	HILIC (7.5–8.5 min)		MALDI-TOF MS, Triple quadrupole and orbitrap MS, enzyme digestion, SEC	[194,239]
*A. catenella*	Rainbow trout gill cell line	F, C	Targeted characterisation of FA and ROS	Crude cell extract						GC-FID, Enzyme detection	[73]
*A. catenella*	Rainbow trout gill cell line	C	Comparative extractions, targeted characterisation of ROS	Comparative extractions in solvent						Enzyme detection, enzyme digestion	[255]
“*A. catenella*” ^a^	Protist (*Skeletonema costatum*)	F	Bioassay guided purification	SPE Amberlite XAD-2 (MeOH)	SPE Sep-pak ODS (MeOH 75%)	SPE Develosil ODS UG-5 (75% MeOH)				HR-FAB MS, NMR, staining	[183]
“*A. catenella*” ^b^	Protists (*Tiarina fusus* and *Polykrikos kofoidii*)	F	Tests on the filtrate							Enzyme digestion, antioxidant addition	[207]
*A. leei*	Whole fish (*Lates calcarifer*)	F	Tests on the filtrate and liquid/liquid partitioning	Liquid partitioning water/hexane/ ethyl acetate	Evaporation and dissolution in DMSO						[174]
*A. minutum*	Protist (*Chaetoceros muelleri*)	F	Bioassay guided purification	SPE C18 (80% MeOH)	HPLC C18, CH_3_CN gradient (17.5–20.6 min)	Evaporation and dissolution in MeOH	HPLC C18, CH_3_CN gradient (8.5–9 min)	Evaporation and dissolution in MeOH		GC-FID, HPLC-MS	[254]
*A. minutum*	Protist (*Chaetoceros muelleri*)	F	Tests on the filtrate								[165]
*A. minutum*	Cytotoxic (Anti-cancer)	C	Bioassay guided purification	Aqueous extraction	SPE Chromabond HR-X (CH_3_CN/H_2_O 7:3)	Ultrafiltration (>10 kDa)				Staining, NMR, HPAEC	[176]
*A. pacificum* ^c^	Protist (*Phaeodactylum tricornutum*)	F	Non-guided purification, characterisation of extracts	Precipitation pH 10–11	Redisolvation of precipitate in 1 N HCl	Dialysis (1 kDa)	Lyophilisation	Dissolution in MeOH		UPLC-MALDI-MS	[195]
*A. tamarense* ^d^	Protists (*Tiarina fusus* and *Polykrikos kofoidii*)	F	Tests on the filtrate							Enzyme digestion, antioxidant addition	[207]
“*A. tamarense*” ^e^	Hemolytic and cytotoxic	F	Bioassay guided purification	Ultrafiltration (>200 kDa)	Ammonium sulfate 100% (precipitate)	Dissolution in water, dialysis	Centrifugation (supernatant)	Lyophilisation and dissolution in PBS	FPLC Superdex-200	SDS-PAGE, staining, amino acid and sugar analysis	[182]
*A. tamutum* ^f^	Copepods and cytotoxic (sea urchin)	C	Targeted characterisation of AA, FA and PUSCA	Crude cell extract						HPLC, GC-MS, NMR	[179]
“*A.* *taylorii*” ^g^	Hemolytic, cytotoxic and shrimp (*Artemia salina*)	F	Bioassay guided purification	Ultrafiltration >10 kDa	Ammonium sulfate 100% (precipitate)	Dissolution in PBS, dialysis	Centrifugation (supernatant)			Staining, enzyme digestion	[180]

The absence of information indicates that no more purification steps were performed. (a) No details (e.g., no strain identifier, no sequence data, and no morphology). (b) In [207] all three strains are reported as *A. tamarense* species complex; two strains are likely to represent *A. catenella* as they originate from the US east coast and produced PSTs. (c) Strain reported as *A. tamarense*. (d) In [207] all three strains are reported as *A. tamarense* species complex, one strain is a true strain of *A. tamarense* [48]. (e) No details (e.g., no strain identifier, no sequence data, and no morphology). (f) Strain of *A. tamutum* reported as *A. tamarense*. (g) No details (e.g., no strain identifier, no sequence data, and no morphology). AA: amino acid, C: cell extract or lysate, CH_3_CN: acetonitrile, DMSO: dimethylsulfoxide, F: filtrate or supernatant, FA: fatty acids, FPLC: fast protein liquid chromatography, GC: gas chromatography, HILIC: hydrophilic interaction ion-chromatography, HPAEC: high-performance anion-exchange, HPLC: high-performance liquid chromatography, MALDITOF-MS: matrix-assisted laser desorption ionisation time-of-flight mass spectrometry, MeOH: methanol, MS: mass spectrometry, NMR: muclear magnetic resonance, PBS: phosphate-buffered saline, PUSCA: polyunsaturated short-chain aldehydes, UPLC: ultra-performance liquid chromatography, ROS: reactive oxygen species, SDS-PAGE: SDS-polyacrylamide gel electrophoresis, SEC: size exclusion chromatography, SPE: solid-phase extraction.

**Table 4 toxins-13-00905-t004:** Chemical characteristics of *Alexandrium* BECs (and unknown intracellular cytotoxins) or hypothesis made on their nature.

		Chemical Properties				Chemical Nature						
Species or Complex	Bioactivity (Target Cell)	Solubility	pH Sensitivity	Thermal Sensitivity	Light Sensitivity	Proteic	Saccharide	Fatty Acids	Reactive Oxygen Species	Molecular Size (Da) or Formula	Other Informations	Study
*A. andersoni*	Cytotoxic (Anti-cancer)	Mid to slightly hydrophilic		Can be frozen (−80 °C)				Detected			Intracellular. Two active fractions. Detection of polar lipids.	[172]
*A. catenella*	Protist (*Rhodomonas salina*)	Amphiphatic	Relatively resistant [3,4,5,6,7,8,9,10,11,12] Lytic activity max at 8	Resistant [−20° to 60], Sensitive (95 °C)	Resistant	No	Detected (Traces)			7–15 kDa	Large non-proteinaceous non-polysaccharide compounds	[194,239]
*A. catenella*	Rainbow trout gill cell line		Resistant [7,8,9]	Resistant [17–85 °C]	Sensitive			PUFA (DHA) detected	Yes		Hypothesis of ROS and peroxidation products	[73]
*A. catenella*	Rainbow trout gill cell line	Hydrophilic							Yes			[255]
“*A. catenella*” ^a^	Protist (*Skeletonema costatum*)	Hydrophobic				No	No	No		C_28_H_49_NO_8_	“Alexandrolide”	[183]
“*A. catenella*” ^b^	Protists (*T. fusus* and *P. kofoidii*)					No			Yes		Hypothesis of ROS and peroxidation products	[207]
*A. leei*	Whole fish (*Lates* *calcarifer*)	Hydrophilic		Resistant to freezing and boiling								[174]
*A. minutum*	Protist (*Chaetoceros muelleri*)	Amphiphatic		Can be frozen (−20 °C)				No		200–700 Da	Molecular size not confirmed by ultrafiltration	[254]
*A. minutum*	Protist (*Chaetoceros muelleri*)			Sensitive (100 °C)								[165]
*A. minutum*	Cytotoxic (Anti-cancer)	Hydrophilic		Can be frozen (−80 °C)		Detected	Detected			>20 kDa	Intracellular. Hypothesis of glycoprotein.	[176]
*A. pacificum* ^c^	Protist (*P. tricornutum*)					Polypeptides detected				1021 Da < size < 1600 Da	Detection of esters and organic acids. Molecular size not confirmed by ultrafiltration.	[195]
*A. tamarense* ^d^	Protists (*T. fusus* and *P. kofoidii*)					Yes			Yes		Hypothesis of ROS and peroxidation products	[207]
“*A. tamarense*” ^e^	Hemolytic and cytotoxic			Can be frozen (−80 °C)		No	Detected			1000 kDa	“AT-toxin”	[182]
*A.* *Tamutum* ^f^	Copepods and cytotoxic (sea urchin)					Amino acid detected		Detected			Absence of PUSCA	[179]
“*A. taylori*” ^g^	Hemolytic, cytotoxic and shrimp (*A. salina*)			Sensitive (100 °C) Not active at 4 °C		Yes				>10 kDa		[180]

The absence of information indicates that no data is available in the article. In the chemical nature, “detected” means that this group has been chemically detected in the extract or active fraction, even though its implication was not confirmed. “Yes” means that the activity could be mitigated with a specific inhibitor. “No” means that the group was not detected, or activity could not be mitigated. (a) No details (e.g., no strain identifier, no sequence data, and no morphology). (b) In [207] all three strains are reported as *A. tamarense* species complex; two strains are likely to represent *A. catenella* as they originate from the US east coast and have PSTs. (c) Strain of *A. pacificum* identified as *A. tamarense*. (d) In [207] all three strains are reported as *A. tamarense* species complex, one strain is a true strain of *A. tamarense* [48]. (e) No details (e.g., no strain identifier, no sequence data, and no morphology). (f) Strain of *A. tamutum* reported as *A. tamarense.* (g) No details (e.g., no strain identifier, no sequence data, and no morphology). AA: amino acid, FA: fatty acids, PUSCA: polyunsaturated short-chain aldehydes, ROS: reactive oxygen species.

**Table 5 toxins-13-00905-t005:** Similarities in bioactivities from *A. minutum* and *A. catenella* strains [59,64,66,67,68,156,157]. The potency is indicated as low, +; moderate, ++; or high, +++. The − sign indicates that no activity was detected. The / sign indicates that the activity was not investigated.

Species	Strain	Type of Toxin	Effects against Protist Competitors or Grazers	Anti-Parasitic Activity	Toxicity to Bivalves
*A. minutum*	CCMI1002	Lytic	+++	+++	+++
*A. minutum*	AM89BM	Lytic + PST	++	++	++
*A. minutum*	Da1257	PST	−	+	−
*A. catenella*	Alex2	Lytic + PST	+++	/	++
*A. catenella*	Alex5	PST	−	/	+

## Data Availability

Not applicable.

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
