# Peer review of "Unknown Extracellular and Bioactive Metabolites of the Genus Alexandrium: A Review of Overlooked Toxins"

_toxins, 2021, doi:10.3390/toxins13120905_

Round 1

Reviewer 1 Report

In this extensive review, the authors synthesize and summarize much of the available bibliography about the genus Alexandrium. The main objective of the work is to highlight the importance of increasing the scarce current knowledge about the so-called bioactive extracellular compounds (BEC), their allelopathic interactions with other marine organisms and with the presence/absence of other known toxins (PSTs or others). The review, therefore, contributes to a better understanding of many aspects (physiology, ecology, toxicology…) of the Alexandrium genera and the dynamics of their blooms. In addition, authors have been an arduous and extensive revision and correction, if applicable, of the nomenclature reported in each of the previous references that they used, justifying in each case the reasons for such changes and highlighting the essential to properly identify and document Alexandrium (and others) species.

In my opinion, this is a good review well-structured and developed and therefore deserves to be published in Toxins. Just some minor comments that must be corrected before its final publication:

  • Rename the title of the paragraph “a toxic genus” since not all species of the Alexandrium genus are toxic.
  • Lines 36. Reference 2 describes the presence of GYMs and SPX in A. peruvianum, not in A. ostenfeldii. Indicate it added “reported as A. peruvianum”.
  • Line 135. Correct “posterorly”
  • Line 248. Correct “gomiodomin A”
  • Table 1. caption. Add the two missing parentheses on the line 272.
  • Table 1. In the second line corresponding to A. andersonii replace “+” by “±” in the column of PSP production according to footnote (b)
  • Line 275. Replace “incl” with the whole word.
  • Line 292. Add the two missing spaces.
  • Table 2. Remove the bold from line 1
  • Table 2 footnotes. Line 333. Add (a) at the beginning of the sentence.
  • Line 425. Correct “phagogotrophy”,
  • Line 433. Correct “overstimation”.
  • Line 566. Correct “unknwown”.
  • Figure 1 caption. Add the meaning of the upward arrow.
  • Line 704. Delete excessive space.
  • Line 716. Add a space between invertebrate and cell.
  • Line 719. Delete excessive space.
  • Line 724-725. Correct “error reference source not found” for the appropriate.
  • Line 729. Correct “prepration”.
  • Table 3. Remove the bold from line 1
  • Line 737. Change symbol [b] by (b).
  • Line 740. Separate “Atamarense”.
  • Line 797. Correct “involvment”.
  • Figure 2. Structure radicals do not look so well. Slightly reduce the zoom of the figure.
  • Line 921. Abbreviation of karlotoxin (KmTx) has not been previously indicated.
  • Line 946. Correct “ressources”.
  • Reference 145. Correct ???

Author Response

Reviewer 1

In this extensive review, the authors synthesize and summarize much of the available bibliography about the genus Alexandrium. The main objective of the work is to highlight the importance of increasing the scarce current knowledge about the so-called bioactive extracellular compounds (BEC), their allelopathic interactions with other marine organisms and with the presence/absence of other known toxins (PSTs or others). The review, therefore, contributes to a better understanding of many aspects (physiology, ecology, toxicology…) of the Alexandrium genera and the dynamics of their blooms. In addition, authors have been an arduous and extensive revision and correction, if applicable, of the nomenclature reported in each of the previous references that they used, justifying in each case the reasons for such changes and highlighting the essential to properly identify and document Alexandrium (and others) species.

In my opinion, this is a good review well-structured and developed and therefore deserves to be published in Toxins. Just some minor comments that must be corrected before its final publication:

  • Rename the title of the paragraph “a toxic genus” since not all species of the Alexandrium genus are toxic.
  • The title has been modified to “1.1. Alexandrium, a genus potentially toxic”
  • Lines 36. Reference 2 describes the presence of GYMs and SPX in A. peruvianum, not in A. ostenfeldii. Indicate it added “reported as A. peruvianum”.
  • It is now indicated: “such as spirolides and/or gymnodimines produced by ostenfeldii (Paulsen) Balech & Tangen ([2], there reported as A. peruvianum)”
  • Line 135. Correct “posterorly”
  • This has been corrected.
  • Line 248. Correct “gomiodomin A”
  • This has been corrected.
  • Table 1. caption. Add the two missing parentheses on the line 272.
  • This has been corrected.
  • Table 1. In the second line corresponding to A. andersonii replace “+” by “±” in the column of PSP production according to footnote (b)
  • This has been corrected.
  • Line 275. Replace “incl” with the whole word.
  • This has been corrected.
  • Line 292. Add the two missing spaces.
  • This has been corrected.
  • Table 2. Remove the bold from line 1
  • This has been corrected.
  • Table 2 footnotes. Line 333. Add (a) at the beginning of the sentence.
  • This has been corrected
  • Line 425. Correct “phagogotrophy”,
  • This has been corrected.
  • Line 433. Correct “overstimation”.
  • This has been corrected.
  • Line 566. Correct “unknwown”.
  • This has been corrected.
  • Figure 1 caption. Add the meaning of the upward arrow.
  • This has been corrected.
  • Line 704. Delete excessive space.
  • This has been corrected.
  • Line 716. Add a space between invertebrate and cell.
  • This has been corrected.
  • Line 719. Delete excessive space.
  • This has been corrected.
  • Line 724-725. Correct “error reference source not found” for the appropriate.
  • This has been corrected.
  • Line 729. Correct “prepration”.
  • This has been corrected.
  • Table 3. Remove the bold from line 1
  • This has been corrected.
  • Line 737. Change symbol [b] by (b).
  • This has been corrected.
  • Line 740. Separate “Atamarense”.
  • This has been corrected.
  • Line 797. Correct “involvment”.
  • This has been corrected.
  • Figure 2. Structure radicals do not look so well. Slightly reduce the zoom of the figure.
  • This figure has been removed and replaced by Figure 1.
  • Line 921. Abbreviation of karlotoxin (KmTx) has not been previously indicated.
  • This has been corrected.
  • Line 946. Correct “ressources”.
  • This has been corrected.
  • Reference 145. Correct ???
  • This has been corrected.

Reviewer 2 Report

 Secondary extracellular metabolites of the genus Alexandrium: a review of overlooked toxins

This is a complete review of known Alexandrium secondary metabolites. The authors have done extensive literature research. This information, without doubt, deserves to be published because it could be a guide for future research.

Nevertheless, some minor details have to be revised.

There are several typos, all marked in the PDF I am attaching, and many italics mistakes.

Please, check the use of commas. Several missing commas make the reading and understanding of the text (some of them I marked).

There are also several double spaces I marked.

Please stick to American or British English. Do not mix.

Please check all the superscripts that are missing.

Some references are missing, like the one from AlgaeBase. Here https://www.algaebase.org/about/, the authors can find the proper way to cite this website.

L220: please clarify. This sentence is not clear nor easy to read.

L248: please display the reference accurately.

Table 1. (and Table 2) Please check the spelling, italics, and grammar. Also, this Table (as in Table 2) is very hard for the eyes. Erase most horizontal bold lines. If the authors want to use lines and the Editor accepts this, please use fine lines. Otherwise, it is hard to read. Also, check capitalization. Probably there is a better way to present this Table  (and Table 2).

Also, please check the footnote. If a Table requires a footnote that is much bigger than the Table, maybe it is time to look for a better Table.

L321, 327: energetic

L324: add commas, otherwise this sentence is not readable.

L351: please add the complete reference here.

L422-423: clarify this sentence

L517-520: The authors always imply that the effects exerted on different species are due to one BEC or toxin but never contemplate the possibility of the mix of different compounds that probably will not affect their own.

L545: The authors reported Crassostrea (Magallana) gigas, which made me check this name change. The authors are correct. However, in the rest of the document, they kept writing C. gigas.

L567: please specify genus and species.

L605: the authors claim that the effects of cultures, cell extracts, or supernatants were not correlated to the PST content of the cells, but they do not mention how this was possible. Please clarify how.

L617: HPLC is not a detection but a separation method. HPLC is not capable of  ”detecting” anything. A detector coupled to the HPLC is needed. The authors also forgot to mention the FLD detector, commonly used for PST detection (after HPLC separation).

L623: please clarify why do authors use quotation marks on ”experiments.”

L643: please explain why toxicity assays on blood cells are not environmentally relevant.

L724: please check the reference administrator.

Table 3: please present this information in a better way to be readable and understandable.

L770-777: The authors claim to be ”most likely A. catenella,” and, without any proof, they still mention the results obtained assuring this was the species. Please, change these sentences in order to clarify this.

L790: please check this number.

Figure 2: this figure is not good. Find a good one or draw it yourselves. There are a myriad of programs for that.

Table 4. Please re-do it so that it is readable and understandable. Probably using a landscape page could help.

Figure 3. The authors claim that a polyhydroxylated chain is hydrophobic, and this cannot be right.

L1012: why the :?

Author Response

Reviewer 2

This is a complete review of known Alexandrium secondary metabolites. The authors have done extensive literature research. This information, without doubt, deserves to be published because it could be a guide for future research.

Nevertheless, some minor details have to be revised.

There are several typos, all marked in the PDF I am attaching, and many italics mistakes.

Thank you for the detailed corrections, they have all been taken into account.

Please, check the use of commas. Several missing commas make the reading and understanding of the text (some of them I marked).

This has been corrected.

There are also several double spaces I marked.

This has been corrected.

Please stick to American or British English. Do not mix.

This has been thoroughly checked and corrected in the whole manuscript.

Please check all the superscripts that are missing.

This has been corrected.

Some references are missing, like the one from AlgaeBase. Here https://www.algaebase.org/about/, the authors can find the proper way to cite this website.

The two missing references were added.

L220: please clarify. This sentence is not clear nor easy to read.

  • This has now been modified: “Later, spirolides were reported to be produced by Alexandrium ostenfeldii [84], the only known species producing ”

L248: please display the reference accurately.

  • This has been corrected.

Table 1. (and Table 2) Please check the spelling, italics, and grammar. Also, this Table (as in Table 2) is very hard for the eyes. Erase most horizontal bold lines. If the authors want to use lines and the Editor accepts this, please use fine lines. Otherwise, it is hard to read. Also, check capitalization. Probably there is a better way to present this Table (and Table 2).

  • Both tables have been modified..

Also, please check the footnote. If a Table requires a footnote that is much bigger than the Table, maybe it is time to look for a better Table.

  • We do agree with the reviewer that here footnotes are large as compared to the table itself. The comments in the footnotes provide additional information that justify why we categorised the production of toxins as “-”, “+” or “±”. Such detailed information could not be indicated in the table because of the place needed, and we did not add it to the main text as “known” toxins are not the main focus of this study. Therefore, we chose to add this information as a footnote for those who are interested.

L321, 327: energetic

  • This has been corrected.

L324: add commas, otherwise this sentence is not readable.

  • This has been corrected.
  • L351: please add the complete reference here.
  • This has been corrected.
  •  

L422-423: clarify this sentence

  • This has now been clarified: “Another additional level of benefit is when organic matter of competitors (either particulate or dissolved) is used by the allelochemical donor species, a “kill and eat your competitor” strategy [203,204].”.

L517-520: The authors always imply that the effects exerted on different species are due to one BEC or toxin but never contemplate the possibility of the mix of different compounds that probably will not affect their own.

  • We agree with the reviewers that effects may also be due to a mix of different effects. We therefore modified the text to make it clear and avoid misinterpretation: “Further comparative analysis with Alexandrium strains with specific bioactivities (e.g., only PST, only lytic, only anti-grazing) or mixed bioactivities are required to assign observed effects to a single group of compounds or to a mixture of known toxins and BEC.”

L545: The authors reported Crassostrea (Magallana) gigas, which made me check this name change. The authors are correct. However, in the rest of the document, they kept writing C. gigas.

  1. gigas has been replace by M. gigas in the whole review.

L567: please specify genus and species.

This has now been added.

L605: the authors claim that the effects of cultures, cell extracts, or supernatants were not correlated to the PST content of the cells, but they do not mention how this was possible. Please clarify how.

This has been modified for more clarity: “The toxicity varied with strains and sample source (i.e. whole culture, cell extracts or supernatant) but was not correlated to the varying PST content of the different strains, and was therefore linked to other group(s) of compounds” .

L617: HPLC is not a detection but a separation method. HPLC is not capable of  ”detecting” anything. A detector coupled to the HPLC is needed. The authors also forgot to mention the FLD detector, commonly used for PST detection (after HPLC separation).

HPLC-FLD is now mentioned.

L623: please clarify why do authors use quotation marks on ”experiments.”

The quotation marks were deleted as they were not relevant.

L643: please explain why toxicity assays on blood cells are not environmentally relevant.

We agree that “environmentally relevant” was slightly misleading and rephrased the sentence:  “It is unlikely that blood cells are directly exposed to the toxins and thus are not the ecological relevant targets, but blood cells of various organisms (including humans) are commonly used in toxicity assays”.

L724: please check the reference administrator.

This has been modified.

Table 3: please present this information in a better way to be readable and understandable.

Tables 3 and 4 were modified for clarity.

L770-777: The authors claim to be ”most likely A. catenella,” and, without any proof, they still mention the results obtained assuring this was the species. Please, change these sentences in order to clarify this.

We do agree with the reviewer that we do not have proof that this strain is A. catenella but as stated in the review, two strains of the three strains likely to represent A. catenella because they originate from the US east coast and have PST. As we can’t be sure of the species, everytime “A. catenella” was cited for this study, quotation marks were used and we added the information (in the text and in the footnote of tables) that it is likely to be A. catenella but that it was not properly investigated.

L790: please check this number.

This was checked and the number is correct according to the reference.

Figure 2: this figure is not good. Find a good one or draw it yourselves. There are a myriad of programs for that.

Figure 2 has been deleted, the structure of Alexandrolide is now displayed in the new Figure 1.

Table 4. Please re-do it so that it is readable and understandable. Probably using a landscape page could help.

Tables 3 and 4 were modified for clarity.

Figure 3. The authors claim that a polyhydroxylated chain is hydrophobic, and this cannot be right.

The Figure 3 was modified accordingly.

L1012: why the :?

This was a mistake, it was removed.

Reviewer 3 Report

After careful revision of the manuscript entitled ‘Secondary extracellular metabolites of the genus Alexandrium: a review of overlooked toxins’ I suggest to totally reorganize the overall review before submission to toxins journal. The importance and the utility to review the data available are not rightly highlighted and/or confusing, I thought to find it in the key contribution but this part has been underestimated by authors. The paragraphs could be shortened to be more understandable (evaluating also if all is necessary to report) and the Table revised to be clearer. Also the Figures need to be improved (see for example Fig. 2).

Author Response

After careful revision of the manuscript entitled ‘Secondary extracellular metabolites of the genus Alexandrium: a review of overlooked toxins’ I suggest to totally reorganize the overall review before submission to toxins journal. The importance and the utility to review the data available are not rightly highlighted and/or confusing, I thought to find it in the key contribution but this part has been underestimated by authors. The paragraphs could be shortened to be more understandable (evaluating also if all is necessary to report) and the Table revised to be clearer. Also the Figures need to be improved (see for example Fig. 2).

General reply: We regret that the reviewer found the importance and utility of this review “confusing”. However, the reviewer did not clearly indicate of what exactly should be improved? However, according to the other reviewer’s comments, a significant amount of work was made to further improve the manuscript: the title of the manuscript was modified to better indicate the focus of the study, the english was revised, and a couple of unclear sentences were modified

specific suggestions

The importance and the utility to review the data available are not rightly highlighted and/or confusing, I thought to find it in the key contribution but this part has been underestimated by authors.

Reply: We now have modified the key contribution of the manuscript: “This review highlights the importance of bioactive extracellular compounds within the genus Alexandrium. We report the wide spectra of activities of these unknown metabolites, their physiological effects on organisms, the available chemical information, and we point out the need to always quantify these activities and underline the need to identify these metabolites.”

the Table revised to be clearer.

Reply: all tables were revised and re-formatted for clarity.

Also the Figures need to be improved (see for example Fig. 2).

Reply: Several reviewers raised concern about Figure 2. For unknown reasons our original figure was not properly included in the version sent to the reviewers, so the quality of Fig. 2 as seen by the reviewers indeed was not acceptable. In any case, figure 2 was now replaced by a new Figure 1 to meet Reviewer 4 recommendation who asked to provide the core chemical structures of the secondary metabolites. We therefore included Alexandrolide along with Saxitoxin, 13-desmethyl spirolide C, Gymnodimine A and Goniodomin A in the same figure.

Moreover, Fig. 3 was slightly modified

We hope that all changes in the revision were sufficient to improve the manuscript and make it clearer.

Reviewer 4 Report

The reviewer has no major concerns about this reviewer manuscript. It can be publishable, but several points need addressing clearly in the revision.

  1. The present title is too abstract, the genus Alexandrium produces a lot of secondary metabolites. The authors should specifically point which group of secondary metabolites are focused in this work.
  2. I strongly suggest that the authors provide the core chemical structures, molecular weight etc. of the mainly discussed secondary metabolites in the manuscript.
  3. Figure 2 needs to be improved, the present one is not an optimal one.

Author Response

Reviewer 4

The reviewer has no major concerns about this reviewer manuscript. It can be publishable, but several points need addressing clearly in the revision.

1. The present title is too abstract, the genus Alexandrium produces a lot of secondary metabolites. The authors should specifically point which group of secondary metabolites are focused in this work.

We do agree with the reviewer that a pletora of secondary metabolites are produced. To better indicate the focus of our study We modified the title to “Unknown extracellular and bioactive metabolites of the genus Alexandrium: a review of overlooked toxins”

2. I strongly suggest that the authors provide the core chemical structures, molecular weight etc. of the mainly discussed secondary metabolites in the manuscript.

The chemical structures and molecular weight of the main toxins are now included in the new Figure 1. Only a few secondary metabolites were given as the classical known phycotoxins were not the core of this review.

3. Figure 2 needs to be improved, the present one is not an optimal one.

Figure 2 has been deleted, the structure of Alexandrolide is now displayed in the new Figure 1.

Round 2

Reviewer 3 Report

The review has been improved enough.